# Copper(I)-nitrene platform for chemoproteomic profiling of methionine

Samrat Sahu [1,5], Benjamin Emenike[1,5], Christian Michel Beusch[2,3], Pritha Bagchi [4], David Ezra Gordon[2] & Monika Raj [1] ✉

Methionine plays a critical role in various biological and cell regulatory processes, making its chemoproteomic profiling indispensable for exploring its functions and potential in protein therapeutics. Building on the principle of rapid oxidation of methionine, we report Copper(I)-Nitrene Platform for robust, and selective labeling of methionine to generate stable sulfonyl sulfimide conjugates under physiological conditions. We demonstrate the versatility of this platform to label methionine in bioactive peptides, intact proteins (6.5-79.5 kDa), and proteins in complex cell lysate mixtures with varying payloads. We discover ligandable proteins and sites harboring hyperreactive methionine within the human proteome. Furthermore, this has been utilized to profile oxidation-sensitive methionine residues, which might increase our understanding of the protective role of methionine in diseases associated with elevated levels of reactive oxygen species. The Copper(I)-Nitrene Platform allows labeling methionine residues in live cancer cells, observing minimal cytotoxic effects and achieving dose-dependent labeling. Confocal imaging further reveals the spatial distribution of modified proteins within the cell membrane, cytoplasm, and nucleus, underscoring the platform's potential in profiling the cellular interactome.

Despite being often post-translationally excised and having a low abundance of proteins (around 2%), methionine plays an important role in various cell-regulatory processes[1–7]. The redox cycle between methionine and methionine sulfoxide provides an excellent antioxidant defense platform against various oxidizing species in cells[8–10]. Studies have suggested that impairment of those functions result in varying diseases, including neurodegeneration, cancer, and cardiovascular diseases[11–13]. The high importance of methionine in regulating biological processes and its implication in diseases makes methionine extremely valuable in chemoproteomics. However, due to the high kinetic reactivity of methionine to form methionine sulfoxide and its poor hydrophilicity most of the methionine residues are found in the interior hydrophobic core, rendering their labeling difficult[1]. Therefore, to date, only a few strategies have been developed for selective modification of methionine, including S-alkylation, sulfimidation using

oxaziridines, and alkylation via α-thiol radicals[14–21]. Out of the three strategies, only oxaziridine has been used for chemoproteomic profiling of methionine because other strategies are limited by the requirement of acidic pH (pH-3), and high excess of probes for the efficient protein labeling[22]. As such, we postulate utilizing a transition-metal catalyzed nitrene pathway as an alternative strategy for methionine labeling (Fig. 1a). This mechanistically different methionine labeling technique would lead to the identification and discovery of ligandable proteins and hyperreactive methionine residues at the active sites that has not been discovered before. In this work, we introduce the Copper(I)-nitrene platform (CuNiP) for the sulfonyl sulfimidation of methionine under mild physiological conditions (Fig. 1b). We demonstrate that the reaction is robust, chemoselective, and exhibits broad substrate scope by efficient modification of a range of bioactive peptides, intact proteins of broad molecular weights

[1]Department of Chemistry, Emory University, Atlanta, GA, USA. [2]Department of Pathology and Laboratory Medicine, Emory University, Atlanta, GA, USA. [3]Department of Surgical Sciences, Uppsala University, Uppsala, Sweden. [4]Department of Biochemistry, Emory University, Atlanta, GA, USA. [5]These authors contributed equally: Samrat Sahu, Benjamin Emenike. ✉e-mail: monika.raj@emory.edu

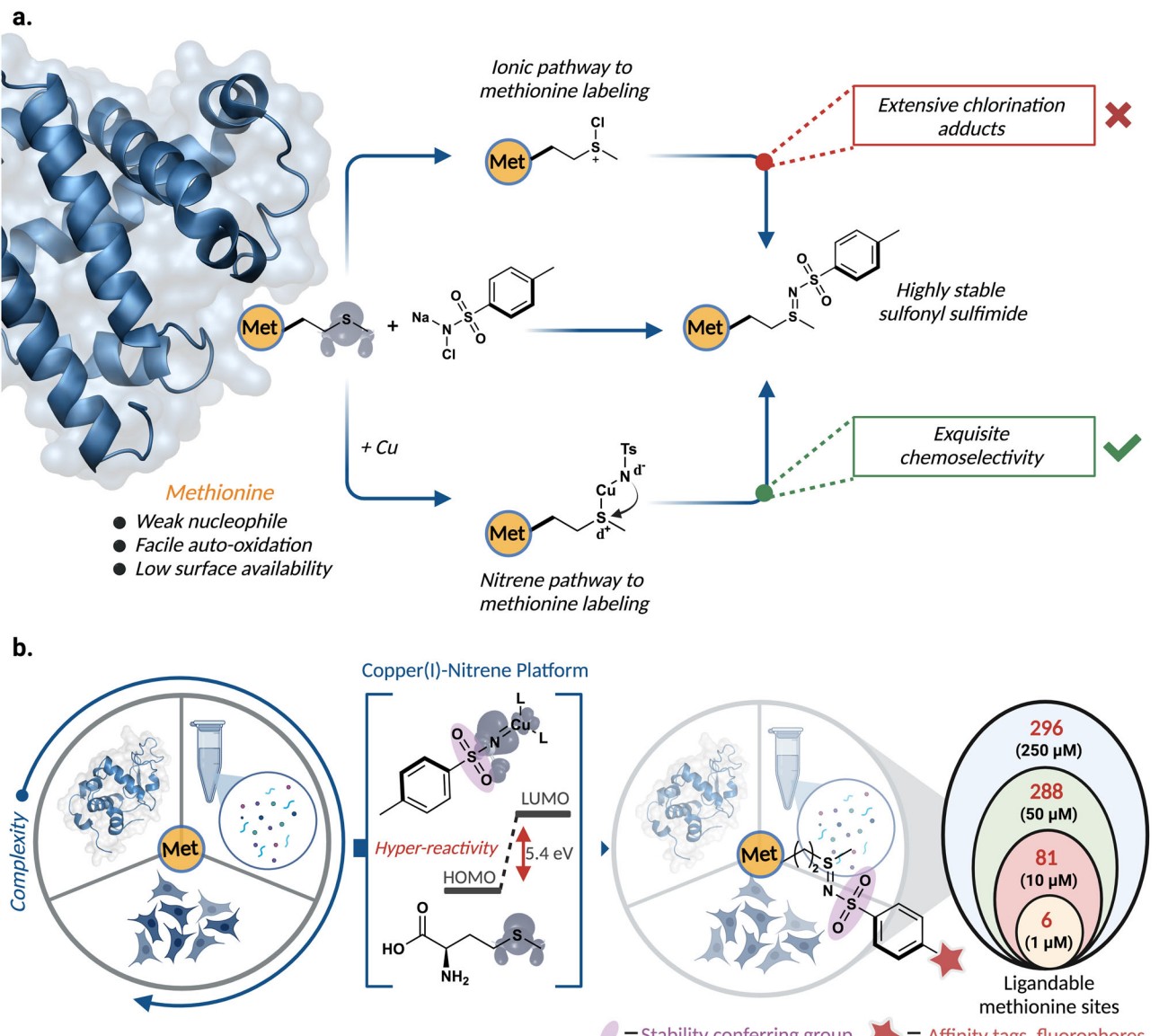

**Fig. 1 | Copper (I) nitrene platform (CuNiP) for selective modification of methionine. a** Plausible reaction pathways for selective modification of methionine including metal-free ionic pathway and metal-catalyzed nitrene pathway with exquisite chemoselectivity for methionine via nitrene-pathway. **b** This work: Copper(I) nitrene platform (CuNiP) for selective modification of methionine for the formation of highly stable sulfonyl sulfimide product and its application in chemoproteomic profiling of human proteome. Figure 1, created with BioRender.com, released under a Creative Commons Attribution-NonCommercial-NoDerivs 4.0 International license" (Agreement number: RQ26NYDQF6).

(6.5–79.5 kDa), complex whole cell lysates and live cells with varying payloads. Most importantly, the resulting sulfonyl sulfimide is extremely stable towards hydrolysis due to the sulfonyl moiety that increases the electron density on nitrogen, thus reducing the electrophilicity of the sulfimide bond towards hydrolysis. The chemoproteomic profiling of human proteome by these mechanistically variable probes led to the discovery of diverse classes of proteins with hyperactive methionine that has not been identified before, thus providing additional ligandable sites (Fig. 1b).

## Results and discussion
### Development of the methionine selective labeling
Initial efforts during the development phase focused on a non-nucleophilic pathway for methionine labeling. This is crucial as it ensures a highly chemoselective method for methionine labeling in the presence of other nucleophilic residues such as Cys, Lys, His, and Tyr.

We started with quantum mechanical studies to assess the intrinsic reactivity of methionine compared to other reactive amino acids. To achieve this, we carried out geometry optimization and natural bond order (NBO) analysis of methionine and other reactive amino acids using density functional theory (DFT) method B3LYP, 6-311 g + +(d,p) basis set, and the solvation model density (SMD) with water as solvent. Our analysis revealed that methionine has a lower HOMO-LUMO (5.7 eV) gap than highly nucleophilic amino acid residues such as lysine (6.4 eV) and cysteine (6.3 eV) (Fig. 2a, Supplementary Fig. 1)[23]. Further analysis of the orbitals showed that the HOMO of methionine is located on the polarizable thioether of methionine. This observation piqued our interest in developing a chemical method that would utilize the polarizable electrons of methionine thioether to achieve the desired modification.

Based on the computational observations, we hypothesized that sulfonyl sulfimidation of methionine could be carried out by *N*-tosyl

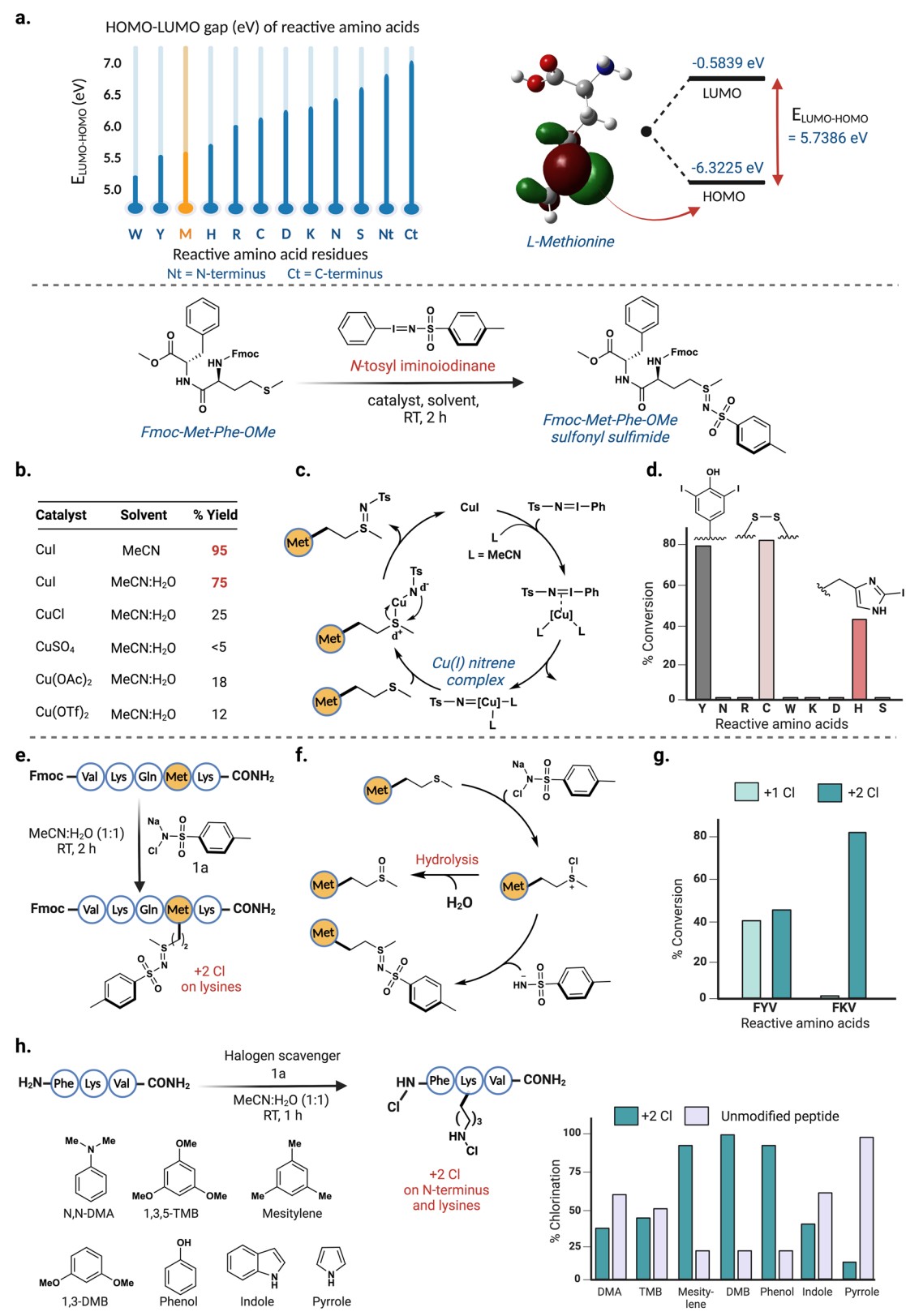

iminoiodinane using copper catalysts through nitrene pathway[24–28]. To optimize the reaction conditions, we carried out the reaction with Fmoc-MF-OMe as a model substrate and screened various Cu(I) and Cu(II) salts such as CuI, CuCl, CuSO₄, Cu(OAc)₂, and Cu(OTf)₂ using Ph-I=N-Ts as sulfimidating agent. The results showed the highest yield with CuI with 95% yield in MeCN and 75% yield in MeCN:H₂O mixture (entries 1 and 2, Fig. 2b, Supplementary Fig. 2). Mechanistically, N-tosyl

iminoiodinane in the presence of Cu(I) generates a Cu-nitrene complex followed by the subsequent release of iodobenzene (PhI). The Cu-nitrene complex further gets trapped by the polarizable thioether moiety of the methionine side chain and generates sulfonyl sulfimide product with methionine (Fig. 2c)[24]. The chemoselectivity studies on tripeptides FXV-CONH₂ with reactive amino acids, where X = Y, N, R, C, W, K, D, H, and S showed bis-iodination of tyrosine (Y), oxidation of

**Fig. 2 | Development of non-nucleophilic chemical approach for the selective labeling of methionine. a** HOMO-LUMO gap of the methionine (5.7 eV) is lower than other reactive nucleophilic amino acids such as lysine (6.4 eV) and cysteine (6.3 eV), suggesting a high kinetic reactivity of methionine. **b** Use of *N*-tosylimi-noiodinane with various copper salts for selective modification of methionine. Reaction with CuI in MeCN gave 95% (red) yield to the sulfonyl sulfimide product, with reduction in yield 75% (red) observed with the introduction of 20% water to the reaction **c** Cu(I)-nitrene mechanism for the formation of sulfonyl sulfimide product with methionine using *N*-tosyl iminoiodinane (Ph-I = N-Ts). **d** Chemoselectivity studies with *N*-tosyl iminoiodinane showed side-reactivity with Tyr (iodination), Cys (disulfide) and His (iodination). **e** Sulfonyl sulfimidation using water-soluble **1a** on Fmoc-VKQMK-CONH₂ shows chlorination on lysine as a side product. **f 1a** carrying out the chlorination reaction via ionic pathway. **g** Chemoselectivity studies of **1a** with FXV-CONH₂ (X = Y and K) showed chlorination of Tyr, Lys, and *N*-terminus. **h** Screening of halogen-trapping reagents for suppressing the chlorination reaction showed that pyrrole is the best halogen-scavenger. Figure 2, created with BioRender.com, released under a Creative Commons Attribution-NonCommercial-NoDerivs 4.0 International license" (Agreement number: OE26NYDGX5).

cysteine (C) to disulfide, and mono-iodination of histidine (H) under the reaction conditions (Fig. 2d, Supplementary Fig. 3). The lower chemoselectivity coupled with the poor solubility of the iminoiodi-nanes in an aqueous medium motivated us to look for different sulfonyl sulfimide alternatives. In this direction, we focused on *N*-chloro-*p*-tolylbenzenesulfonamide sodium salt (chloramine-T), **1a** as a sulfonyl sulfimidating agent for labeling methionine[29,30].

Using **1a** as a sulfimidating reagent, we carried out the reaction on a model peptide Fmoc-VKQMK-CONH₂ in MeCN:H₂O (1:1) and observed the formation of ~3:1 sulfonyl sulfimide product:sulfoxide on Met along with the considerable amount of *N*-chlorination of the lysine residues (Fig. 2e, Supplementary Fig. 4)[31,32]. The *N*-chlorination of lysine stems from the fact that **1a** operates via an ionic mechanism and acts as a chlorinating agent, in which it first chlorinates the thioether to give S⁺-Cl intermediate (Fig. 2f). The attack of tosylamide (TsNH⁻) or hydrolysis of S⁺-Cl intermediate generates the corresponding sulfonyl sulfimide product or sulfoxide, respectively (Fig. 2f). We characterized the formation of sulfonyl sulfimide product by synthesizing it on methionine ester, Fmoc-Met-OMe by using NMR and HRMS (Supplementary Fig. 4). A brief chemoselectivity study on tripeptides FXV-CONH₂ (X = reactive amino acids) showed that various nucleophilic residues such as tyrosine (Y), lysine (K), and *N*-terminus underwent chlorination within 1 h (Fig. 2g, Supplementary Fig. 5).

To suppress this unwanted chlorination, we introduced various halogen-trapping reagents such as *N,N*-dimethylaniline (*N,N*-DMA), 1,3,5-trimethoxybenzene (1,3,5-TMB), mesitylene, 1,3-dimethoxy benzene (1,3DMB), phenol, and heteroaromatics such as indole and pyrrole[33]. The screening of these halogen scavengers on a model peptide FKV-CONH₂ showed that the superstoichiometric addition of pyrrole suppresses the *N*-chlorination significantly as the starting peptide could be isolated in 88% yield after 1 h (Fig. 2h, Supplementary Fig. 6). However, the product:sulfoxide ratio could not be improved further. In addition, the poor solubility of pyrrole in aqueous media and the nonapplicability of this protocol on proteins motivated us to look for suitable alternatives.

To circumvent the chlorination issues, we hypothesized that switching from an ionic to nitrene mechanism would have two-fold benefits in the product outcome[34–38]. First, under the nitrene pathway, **1a** will release NaCl as the sole by-product that will solve the unwanted chlorination as observed previously (Fig. 3a). Second, since the nitrene pathway does not form any long-lived ionic intermediate, sulfoxide formation will be suppressed even in presence of water as a co-solvent. To achieve the desired switch of mechanism, we utilized a stoichiometric amount of transition-metal salts (MX) to generate a metal-nitrene complex in-situ. The stoichiometric amount of the metal salt was necessary to force the reaction to undergo exclusively through the nitrene mechanism. Using this hypothesis, we screened various 3d and 4d-row transition metals on **1a** for in-situ generation of metal-nitrene complexes and tested them for the modification of a model substrate Fmoc-VKQMK-CONH₂ (Fig. 3b, Supplementary Fig. 7). We found that mainly Cu(I)-salts, and in particular CuBr was most effective compared to other Fe- and Pd-based catalysts, delivering the sulfonyl sulfimide product in 88% yield (Fig. 3b, Supplementary Fig. 7). The computational evaluations of the energy parameters of the copper-nitrene

complex showed that the coordination of acetonitrile as a ligand is more energetically favorable (22 kcal/mol) than the copper-nitrene complex without any ligand (Supplementary Fig. 8). These calculations suggested that acetonitrile (MeCN) used in the reaction could be involved in stabilizing the in-situ generated Cu-nitrene species. Based on this principle, we screened external mono and bidentate ligands such as pyridine, 2,2'-bipyridine, and 1,10-phenanthroline but did not observe any increase in the conversion of sulfonyl sulfimidated product as analyzed by HPLC and MS (Supplementary Fig. 9). To further gain mechanistic insights into how the ligand-bound copper-nitrene complex reacts with methionine, we evaluated the interaction energy of the HOMO of methionine reacting with the LUMO of the MeCN-bound copper-nitrene complex (Fig. 3c, Supplementary Fig. 10).

The energy obtained for this interaction was significantly lower ($\Delta eV_1 = 5.40$ eV) than the interaction of the HOMO of MeCN-bound copper-nitrene complex with LUMO of methionine ($\Delta eV_2 = 5.97$ eV), thus confirming that the lone pair of electrons in the HOMO of methionine attacks the LUMO orbital on the copper of the copper-nitrene complex (Fig. 3c)[39].

To ascertain the chemoselectivity of CuNiP, we computationally evaluated the energetics of MeCN-bound copper-nitrene complex with other reactive amino acid residues such as W, Y, H, R, K, C, N, N-terminus (Nt), S, C-terminus (Ct), and D. From our computational results, we observed that the energy gap between the HOMO of methionine and the LUMO of the MeCN-bound copper-nitrene complex is the lowest among all the reactive amino acid residues except for tryptophan (Figs. 3c and d, Supplementary Fig. 11). The experimental evaluations using a nonapeptide Fmoc-KQYWCREHS-CONH₂ containing all the reactive amino acid residues with the exception of methionine further validated the computational prediction (Fig. 3e, Supplementary Fig. 12). No modification of the peptide was observed even after 24 h and the unreacted peptide was isolated in >90% yield, thus confirming the exquisite selectivity of the CuNiP towards methionine residue (Fig. 3e). Evaluation of CuNiP reaction on dec-apeptide Fmoc-KQYWCRMEHS-CONH₂ containing all the reactive amino acid residues in addition to methionine led to a single mod-ification of the methionine residue in 79% yield, further validating the exquisite chemoselective nature of the CuNiP reaction. While the results obtained from calculations suggest a plausible reactivity of tryptophan with the MeCN-bound copper-nitrene complex, we hypo-thesize that the high resonance stabilization and delocalization of the pi electrons in the HOMO of tryptophan might be responsible for the observed lack of reactivity (Supplementary Fig. 12).

## Stability of sulfonyl sulfimide adducts
Another important feature required for chemoproteomic application is the stability of the resulting conjugates. We began our studies by measuring the rate of decomposition of sulfonyl sulfimide conjugate of Ac-Met-OMe with chloramine-T to the corresponding sulfoxide at 37 °C in MeOD using NMR at regular interval of times. The results indi-cated the exceptionally high stability of the sulfonyl sulfimide con-jugate with no detectable decomposition even after 7 days (Fig. 3f, Supplementary Fig. 13). To gain further insight, we carried out NBO analysis on sulfonyl sulfimides by DFT calculations (Fig. 3g,

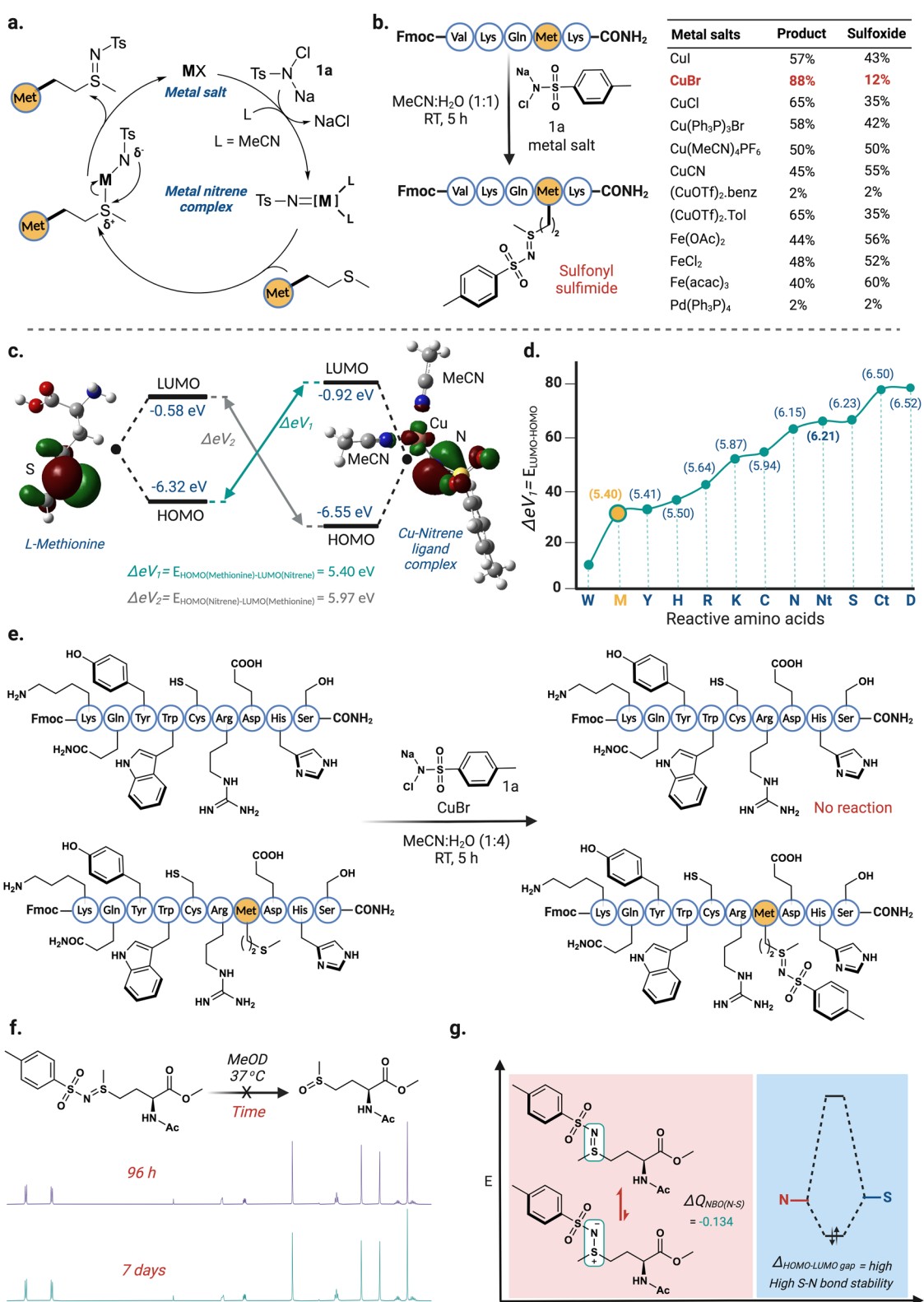

| Metal salts | Product | Sulfoxide |
|---|---|---|
| CuI | 57% | 43% |
| **CuBr** | **88%** | **12%** |
| CuCl | 65% | 35% |
| Cu(Ph₃P)₃Br | 58% | 42% |
| Cu(MeCN)₄PF₆ | 50% | 50% |
| CuCN | 45% | 55% |
| (CuOTf)₂.benz | 2% | 2% |
| (CuOTf)₂.Tol | 65% | 35% |
| Fe(OAc)₂ | 44% | 56% |
| FeCl₂ | 48% | 52% |
| Fe(acac)₃ | 40% | 60% |
| Pd(Ph₃P)₄ | 2% | 2% |

Supplementary Fig. 13). The calculations clearly showed the donation of electron density from nitrene nitrogen to the sulfimide sulfur atom thus decreasing the +ve charge density with a $Q_{NBO(N-S)}$ of (−0.134), hence exhibiting predominantly S-N double bond character.

Consequently, it is plausible to hypothesize a higher HOMO-LUMO gap for sulfonyl sulfimides, hence the observed high hydrolytic stability. To experimentally validate our hypothesis, we carried out stability studies on sulfonyl sulfimide conjugates of Fmoc-MF-OMe at

varying temperatures (RT, 60 °C) and pH conditions (pH ~3–10) for 24 h. No decomposition of the sulfonyl sulfimide conjugate was observed under the reaction conditions (Supplementary Fig. 14).

## Exploration of chloramine-T analogs for selective modification of methionine on myoglobin

To study the effects of substituents on the chloramine-T in the efficiency of Met labeling, we screened chloramine-T analogs containing

**Fig. 3 | Development of CuNiP for sulfonyl sulfimidation of methionine.**
**a** Plausible mechanism for the formation of sulfonyl sulfimide product with methionine via Cu-nitrene pathway. NaCl is the only byproduct of this reaction pathway. **b** Screening of varying metal salts for the formation of sulfonyl sulfimide conjugate with a peptide Fmoc-VKQMK-CONH₂ shows CuBr is the most effective metal salt with 88% product to 12% sulfoxide. The optimal condition with CuBr is highlighted in red. **c** Computational analysis of the interaction energy of HOMO of methionine and LUMO of MeCN-bound Cu-nitrene complex clearly shows the flow of electrons from methionine to nitrene. **d** Calculated HOMO-LUMO gap between HOMO of reactive amino acids and LUMO of MeCN-bound Cu-nitrene species clearly shows the high reactivity and selectivity towards methionine.

**e** Chemoselectivity studies with peptides Fmoc-KQYWCREHS-CONH₂ and Fmoc-KQYWCRMEHS-CONH₂ using chloramine-T and CuBr via Cu-nitrene pathway showed high selectivity for Met. **f** Stability studies of sulfonyl sulfimide towards hydrolysis as analyzed by NMR, showed no detectable decomposition to sulfoxide after 7 days. **g** Computational studies showed high hydrolytic stability of sulfonyl sulfimide due to the predominant double-bond character resulting from the high electron-density on nitrogen of sulfonyl sulfimide thus making it less electrophilic towards hydrolysis. Figure 3, created with BioRender.com, released under a Creative Commons Attribution-NonCommercial-NoDerivs 4.0 International license" (Agreement number: SS26NYD4Q8).

both electron-donating (**1a**; -Me, **1b**; -OMe) and electron-withdrawing groups (**1c**; -Cl, **1d**; -COOH, **1e**; -NO₂, **1f**; 3,5-di-F, **1g**; -CF₃) for the modification of methionine containing pentapeptide Fmoc-VKQMK (Fig. 4a, Supplementary Fig. 15). In general, we observed that probes having electron-donating groups (Me or OMe) gave better product:sulfoxide ratio (**2a**; -Me, 87:13; **2b**; -OMe, 72:28) compared to their electron-withdrawing counterparts (**2c**; -Cl, 66:34; **2d**; -COOH, 38:62; **2e**; -NO₂, 37:63; **2f**; 3,5-di-F, 24:76; **2g**; -CF₃, 54:46). To further evaluate the electronic effect on proteins, we treated myoglobin[40] with 10 equivalence of chloramine-T analogs (**1a-1g**) and 10 equivalence of CuBr in MeCN:H₂O (1:4) for 2 hours. Interestingly. we observed similar trends when CuNiP reaction was applied on myoglobin, with electron-donating (-Me, -OMe) giving better conversion to modified myoglobin (**2a**; -Me, 67% **2b**; -OMe, 44%) compared to their electron-withdrawing counterparts (**2c**; -Cl ~ 24%; **2d**; -COOH, ~ 35%; **2e**; -NO₂, ~30%; **2f**; 3,5-di-F, ~ 32%; **2g**; -CF₃, ~ 42%) (Fig. 4b, Supplementary Fig. 16). The MS/MS data of the modified myoglobin showed labeling ratio of methionine sites M131:M55 as 3:1. To gain mechanistic insight into the experimental observations, we carried out Mulliken population and a Natural Bond Orbital (NBO) analysis (Fig. 4c). The results from this analysis showed that the electron density on the nitrogen ($Q_N$) of the copper-nitrene complex increases when an electron-donating groups are present (**1a**; $Q_{N-Mulliken} = -0.230$, **1b**; $Q_{N-Mulliken} = -0.242$), therefore increasing the partial negative charge (Fig. 4c, Supplementary Fig. 17). This increase in electron density enables the capture of the methionine-Cu-nitrene sulfonium complex to form the sulfonyl sulfimide product more efficiently. Consequently, a decreased electron density on the nitrogen ($Q_N$) of the copper-nitrene complex was observed with chloramine-T containing electron-withdrawing groups (**1c**; $Q_{N-Mulliken} = -0.168$, **1e**; $Q_{N-Mulliken} = -0.160$, **1f**; $Q_{N-Mulliken} = -0.154$ and **1g**; $Q_{N-Mulliken} = -0.131$). A similar trend was observed for the NBO charge ($Q_N$) analysis. To evaluate if the sulfur atom of methionine becomes electrophilic upon reaction with CuNiP probe, we optimized the geometries and calculated the electrostatic potential (ESP) maps of methionine and Met-Cu-Nitrene sulfonium complex. Analysis of the ESP map clearly shows the partial electrophilic nature of the sulfur atom in Met-Cu-Nitrene sulfonium complex (−CH₃ analog, **1a**) as compared to unreacted methionine (Supplementary Fig. 17). Furthermore, the electron density on Cu in the nitrene complex ($Q_{Cu}$) decreased when electron-withdrawing groups were present, thus suggesting that the initial reaction with methionine to form the methionine-Cu-nitrene sulfonium complex is faster in electron-withdrawing probes but the subsequent trapping by the nitrene nitrogen is less favorable, thus the increased formation of sulfoxide was observed with electron-withdrawing probes as compared to electron-releasing probes (Fig. 4c, Supplementary Fig. 17).

## Substrate scope of CuNiP for labeling methionine in proteins
With the optimized conditions for the selective modification of peptides and myoglobin, we next applied CuNiP for the modification of intact proteins of varying molecular weights (6.5 kDa-79.5 kDa) with different three-dimensional architectures (Fig. 5). We explored CuNiP for the modification of Aprotinin, a serine protease inhibitor[41],

containing one methionine residue was modified exclusively without any concomitant oxidation using **1a**, delivering the labeled sulfonyl sulfimide Aprotinin conjugate in 90% conversion (Fig. 5, Supplementary Fig. 18). Despite having only moderate surface accessability, the N-terminal methionine of Ubiquitin generated sulfonyl sulfimide Ubiquitin conjugate in 42% conversion with chloramine-T **1a** under CuNiP optimized conditions (Fig. 5, Supplementary Fig. 19). Ribonuclease A, an endoribonucleases that cleaves the phosphodiester bonds of single-strand RNA after pyrimidine nucleotides[42], was modified with **1a** to give sulfonyl sulfimide products with 50% of +1 modification and 22% +2 modification. The MS/MS analysis showed the sites of sulfimidation with the ratio of M13:M79 as 2.3:1 (Fig. 5, Supplementary Fig. 20). Notably, the results showed a higher preference for labeling M13 site in the presence of four methionines on Ribonuclease A. Next, we attempted the reaction on three other proteins containing two methionine residues such as lysozyme egg-white (14.3 kDa), human lysozyme (14.6 kDa), and α-chymotrysinogen A (25.6 kDa) but surprisingly we did not observe any modification to sulfonyl sulfimide and sulfoxidation product even in the presence of the excess amounts of **1a** probe (up to 50 equiv). We hypothesized that this is probably due to the inaccessibility of the buried methionine residues. To further validate this hypothesis, we denatured these proteins and treated each with **1a** under optimized conditions. Gratifyingly after denaturation, we observed the labeling of methionine to sulfonyl sulfimide with **1a** in all the cases. The MS/MS analysis showed the labeling at M105 site for lysozyme egg-white, M29 site for human lysozyme, and M192 site for α-chymotrypsinogen A (Fig. 5, Supplementary Fig. 21-23). Inspired by the efficiency of CuNiP, we next carried out modification on high molecular weight proteins with complex three-dimensional structures. Carbonic anhydrase, a zinc-containing enzyme that plays a key role in various physiological functions[43], underwent modification with **1a** under physiological conditions resulting in the labeling of only two methionines out of three in the ratio M221:M239 (1.5:1) (Fig. 5, Supplementary Fig. 24). Similarly, wild-type creatine kinase (42.9 kDa) showed labeling of three methionine sites M206:M271:M178 in the ratio (11:2:1) out of 10 Met, whereas bovine serum albumin (66 kDa) showed exclusive labeling at one site M87 out of 4 Met as analyzed by MS/MS (Fig. 5, Supplementary Fig. 25-26). Finally, apo-transferrin (79.5 kDa), containing 9 methionine residues showed reactivity at the M483 and M518 site (M483:M518 labeling ratio 3.3:1) using **1a** (Fig. 5, Supplementary Fig. 27). The ability of CuNiP to selectively modify specific methionine sites among several others corroborates with the chemoproteomics principal where hyperreactive methionines undergo selective labeling based on the microenvironment. This shows the remarkable ability of this platform to detect hyperreactive Met in a proteome.

## CuNiP for installation of payloads in proteins and peptides
Based on the high efficiency of CuNiP in the selective labeling of Met on proteins, we envisioned utilizing CuNiP for installing affinity tags onto a protein of interest at methionine sites, serving as a linchpin for functionalization with various payloads. To achieve this, we screened probes containing various affinity tags such as alkynes (**1h-1i**) with

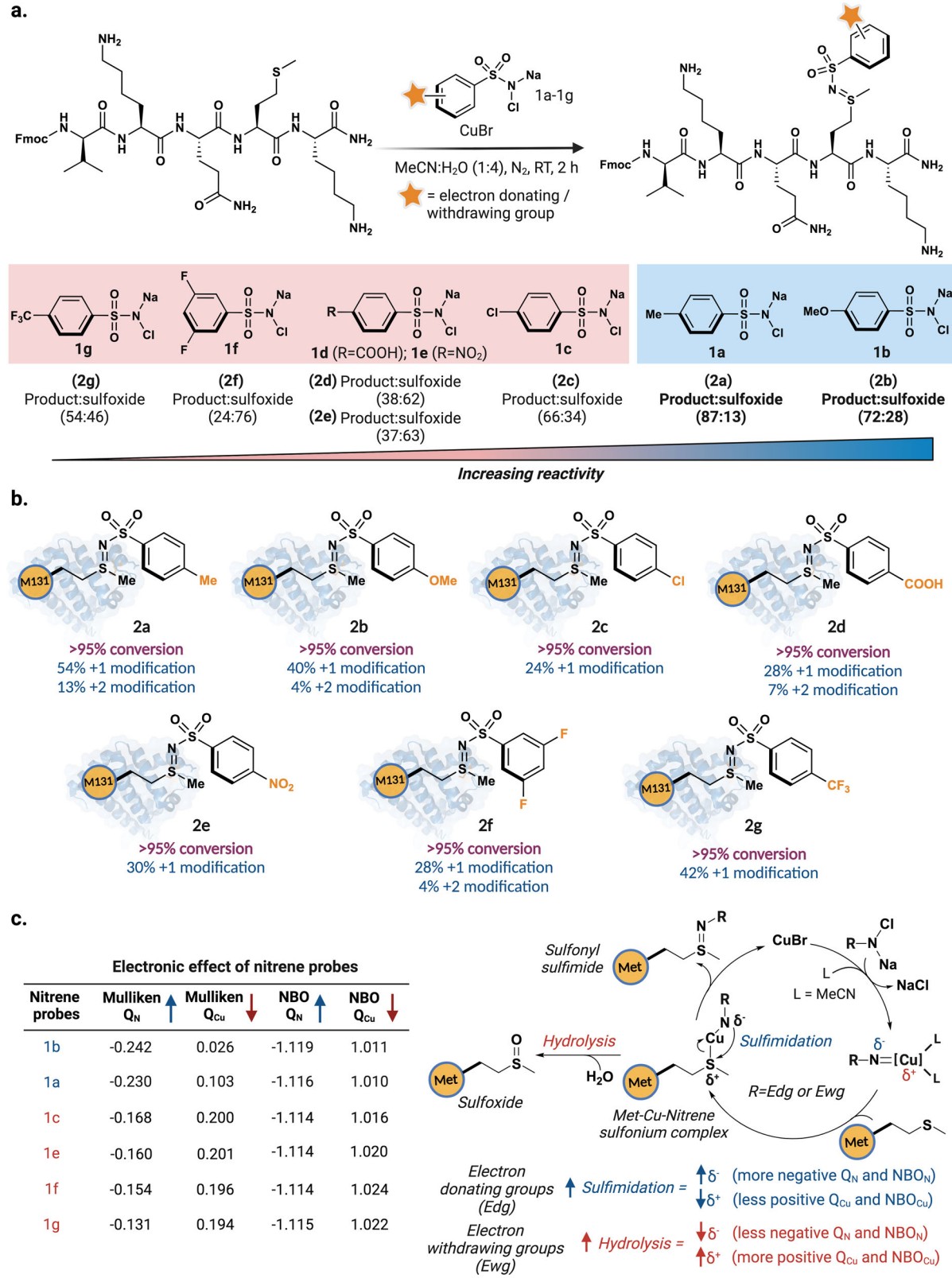

**Fig. 4 | Chemoselective labeling of myoglobin using chloramine-T analogs.**
**a** Labeling of methionine containing pentapeptide Fmoc-VKQMK with various chloramine-T analogs containing electron withdrawing and electron releasing groups using CuNiP. Electron-donating groups (Me or OMe) gave better product:sulfoxide ratio (**2a**; Me, 87:13, **2b**; OMe, 72:28) compared to their electron-withdrawing counterparts (**2c**; Cl, 66:34; **2d**; -COOH, 38:62; **2e**; -NO₂, 37:63; **2f**; 3,5-di-F, 24:76; **2g**; -CF₃, 54:46). **b** Screening of electron-donating and electron-withdrawing chloramine-T analogs on myoglobin. **c** Calculated Mulliken population analysis and Natural Bond Orbital (NBO) analysis on various chloramine-T probes showing electron donating groups (blue) exhibit high efficiency for modifying methionine to sulfonyl sulfimide conjugate than electron withdrawing groups (red), due to an increased electron density on nitrene nitrogen followed by sulfimidation. Figure 4, created with BioRender.com, released under a Creative Commons Attribution-NonCommercial-NoDerivs 4.0 International license" (Agreement number: NZ26NYBOZJ).

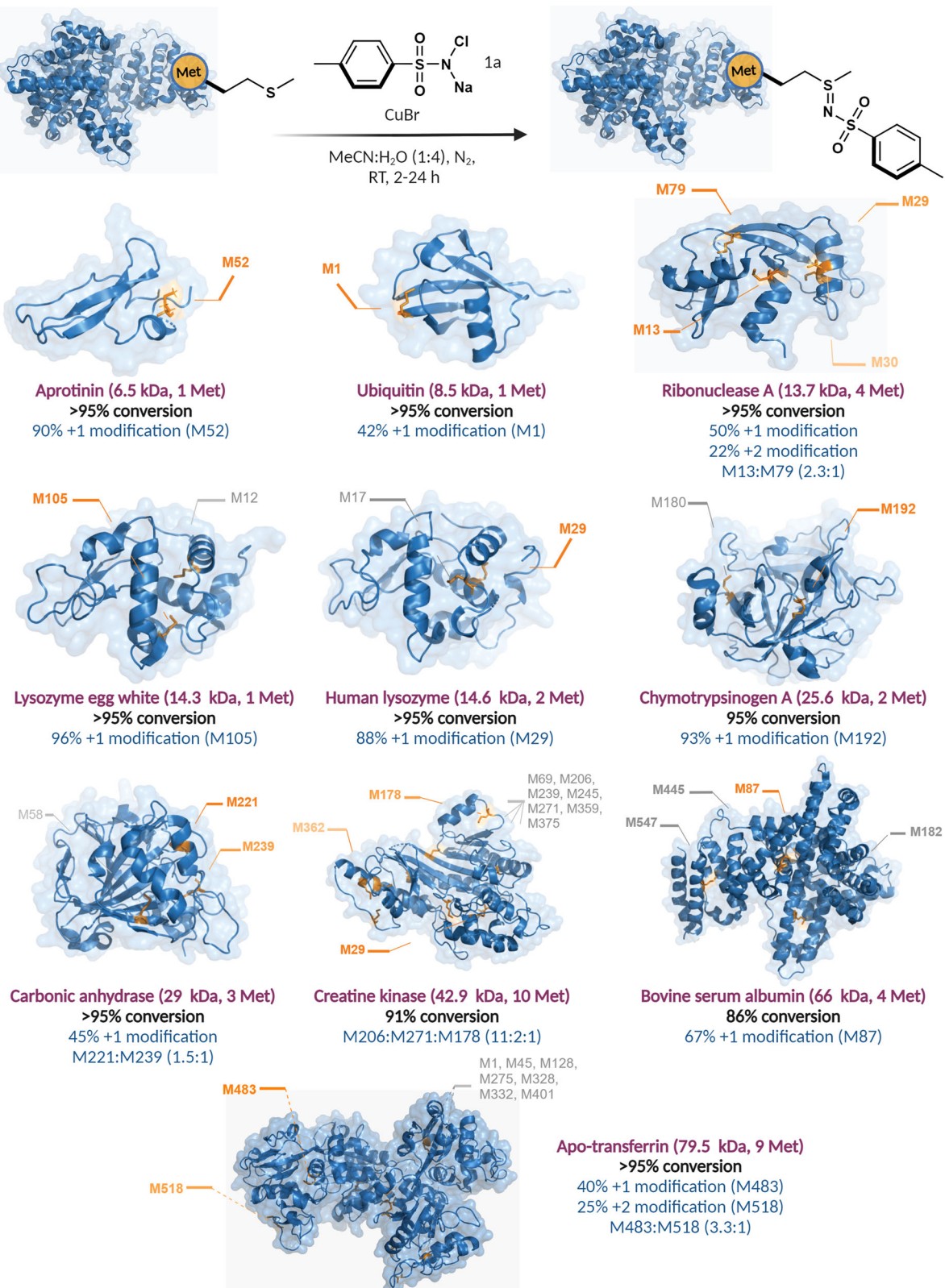

**Fig. 5 | Application of CuNiP for the selective labeling of various intact proteins (6.5-79.5 kDa) using 1a.** The % conversion to modified proteins was analyzed by MS and the site of modification is determined by MS/MS analysis. Ubiquitin reaction was performed in MeCN:0.01 M citrate buffer (1:4) as a solvent. For denatured lysozyme egg-white, lysozyme human, and α-chymotrpsinogen A, reactions were performed in MeCN:H₂O (3:1) as solvent. Figure 5, created with BioRender.com, released under a Creative Commons Attribution-NonCommercial-NoDerivs 4.0 International license" (Agreement number: IL26NYB6ZR).

varying linker lengths and masked aldehyde, **1j** for the selective labeling of myoglobin (Fig. 6a, Supplementary Fig. 28). The results showed the efficient integration of these affinity tags into myoglobin (**2h-2j**) with the highest conversion of methionine to sulfonyl sulfimide conjugate **2i** observed with probe **1i** (47%; 44% +1 mod., 3% +2 mod.) (Fig. 6a, Supplementary Fig. 28). The MS/MS analysis of the modified myoglobin **2h** with affinity tagged-chloramine T **1h** showed site-selective labeling at M131 further confirming that the reaction takes place at the same methionine residue independent of the nature of substituents on the probe (Fig. 6b, Supplementary Fig. 28). This site-selectivity regardless of the nature of substituents on the probe further confirms that hyperreactive Met are more prone to react by CuNIP. Furthermore, the circular dichroism (CD) studies showed no change in the tertiary structure of sulfonyl sulfimide-modified myoglobin as compared to the unmodified myoglobin (Fig. 6c, Supplementary Fig. 29), thus highlighting the ability of CuNiP to modify proteins efficiently and selectively without altering their tertiary structures.

Using the best affinity handle probe **1i**, we next applied the CuNiP to selectively label bioactive peptides (Fig. 6d, Supplementary Fig. 30). Adrenomedullin, a vasodilator known to be associated with pheochromocytoma[44] was efficiently coupled with alkyne containing probe **1i** to generate sulfonyl sulfimidated conjugate with Met, in 65% conversion (Fig. 6d, Supplementary Fig. 30). Tetracosactide acetate, a synthetically derived subunit of the endogenous peptide pituitary hormone adrenocorticotropic hormone (ACTH)[45] reacted effortlessly with **1i** to deliver the conjugated sulfonyl sulfimide product in 65% conversion. Similarly, other bioactive peptides such as aviptadil acetate (injectable synthetic formulation of human vasoactive intestinal peptide)[46] and α-endorphin (neurotransmitter)[47] were effortlessly modified with the alkyne probe **1i**, providing corresponding sulfonyl sulfimide Met conjugates in 65-70% conversions (Fig. 6d, Supplementary Fig. 30).

## CuNiP reaction for installation of fluorophores on proteins in complex cell lysate mixture

Next, we demonstrated the applicability of CuNiP for the selective labeling of native proteins with a fluorophore. We incubated apo-transferrin, bovine serum albumin (BSA), creatine kinase (CK), and myoglobin with alkyne analog **1i** under optimized conditions, followed by the reaction of the sulfonyl sulfimide conjugate of proteins with Cy5 azide using the copper-catalyzed alkyne-azide cycloaddition reaction (CuAAC). The resulting fluorophore-attached proteins were analyzed by SDS-PAGE using in-gel fluorescence (Fig. 7a, Supplementary Fig. 31). The results clearly showed the labeling of all the proteins with fluorophores in the presence of all the three reaction components, CuBr, **1i** and azide fluorophore (lane 3, Fig. 7a, Supplementary Fig. 31). No fluorophore labeling was observed in the absence of any one of the components (lanes 1 and 2, Fig. 7a).

Next, we utilized CuNiP for labeling proteome in a complex cell lysate mixture (Fig. 7b, Supplementary Fig. 32) by incubating breast cancer cell (T-47D) lysate with alkyne containing probe **1i** and CuBr at different doses (50-200 μM) for 2 h followed by attaching Cy5 azide through CuAAC. In-gel fluorescence analysis clearly showed a dose-dependent labeling of Met on proteins in a complex cell lysate mixture (lanes 3-6, Fig. 7b, Supplementary Fig. 32). No labeling was observed in the control experiments in the absence of any one of the components (lanes 1-2, Fig. 7b, Supplementary Fig. 32).

## Chemoproteomic identification of functional methionines in a lysate

Identifying ligandable methionine sites through activity-based protein profiling (ABPP) can revolutionize drug target discovery and the elucidation of protein functions. To begin with, we modified T-47D cell lysates with 150 μM of **1a**, 150 μM CuBr, and 5% acetonitrile, followed by tryptic digestion of proteins. Proteomics analysis of digested lysate identified 623 unique peptides corresponding to 152 proteins (Supplementary Fig. 33, Supplementary data 1). With this result, we proceeded to evaluate the dose-dependent labeling of T-47D cell lysates with alkyne-containing probe **1i**. Lysate samples were treated with 1 μM (super low-dose), 10 μM (low-dose), 50 μM (medium-dose), and 250 μM (high-dose) of probe **1i** for 2 h, followed by tryptic digestion of samples (Fig. 8a, Supplementary Fig. 34). Proteomics analysis of digested lysate identified dose-dependent identification of unique methionine labeled peptides (6 at 1 μM; 118 at 10 μM, 521 at 50 μM, 548 at 250 μM). The amino acid sequence logo was generated using the Plogo map tool[48]. We do not attribute the underrepresentation of K/R residues to cleavage specificity, given that trypsin is generally unaffected by post-translational modifications of amino acids, as observed in kinase motif analysis. Furthermore, we only examined peptides with at least 4 residues upstream and downstream of modified methionine sites, thus eliminating the potential artificial underrepresentation of lysine or arginine resulting from trypsin cleavage.

Heat map analysis of modified methionine sites clearly showed minimal modifications at super low concentrations (1-10 μM) and significant modifications at higher concentrations (50-250 μM). Interestingly, heatmap analysis of modified peptides identified nine proteins within cluster 11 possessing hyper-reactive methionine sites, with modified proteins involved in enzymatic, regulatory, or structural processes, thus serving as potential ligandable target sites for CuNiP (Fig. 8b, Supplementary Fig. 34). Furthermore, extensive GO analysis of all modified proteins exhibited a substantial functional categorization across various classes, with a majority being enzymes and regulatory proteins (Fig. 8c, Supplementary Fig. 34). Solvent accessible surface area (SASA) studies of 200 randomized methionine modified sites from high-dose samples showed a significant modification of buried methionine sites (>74% buried), thus highlighting the potential of CuNiP to label methionine residues irrespective of the spatial location of the residue on a protein (Supplementary Fig. 34). Our sequence motif analysis of the labeled methionines (10 μM dose) showcased an overrepresentation of negatively charged and underrepresentation of positively charged or aromatic residues (Fig. 8d, Supplementary Fig. 34). This is in line with the previous reports showcasing the absence of positively charged and aromatic residues near the oxidation prone methionine residues[49,50]. Notably, CuNiP platform tends to emphasize the influence of local amino acid environments on the reactivity. By understanding this nuanced interplay, this can enable us to better predict reactive sites in proteins, thereby guiding the design of potent covalent modulators. Lastly, we demonstrated the ability of CuNiP to profile oxidation-sensitive methionine within the human proteome. This is particularly important as methionine often serves as a direct scavenger of reactive oxygen species (ROS), often protecting cells from oxidative stress[51]. To profile oxidative sensitive methionine residues with our CuNiP platform, we incubated T47D cell lysate with increasing concentrations of hydrogen peroxide (0.5-2 mM) for 1 h followed by incubation with 250 μM of **1i** and CuBr for 2 h. Click reaction with Cy5 azide showed a dose-dependent decrease in fluorescence intensity as hydrogen peroxide concentration increased, with the highest fluorescent intensity observed for control sample without hydrogen peroxide treatment (Fig. 8e, Supplementary Fig. 35). Proteomics analysis further corroborates this as we observed a dose-dependent decrease in the number of **1i** modified peptides as hydrogen peroxide concentration increased (Fig. 8f, Supplementary Fig. 35). Interestingly, analysis of the impact of hydrogen peroxide on **1i** modified peptides found in control samples as well as in $H_2O_2$ treated samples, further showed a decrease in intensity of modified sites (356 peptides-control; 267 peptides-0.5 mM; 178 peptides- 1 mM; 115 peptides- 2 mM) (Fig. 8g, Supplementary Fig. 35), with 86 sites exclusively found only in control but not in any of the $H_2O_2$ samples (Supplementary Fig. 35). GO analysis of these proteins containing oxidation sensitive methionine sites showed significant enrichment of nucleic

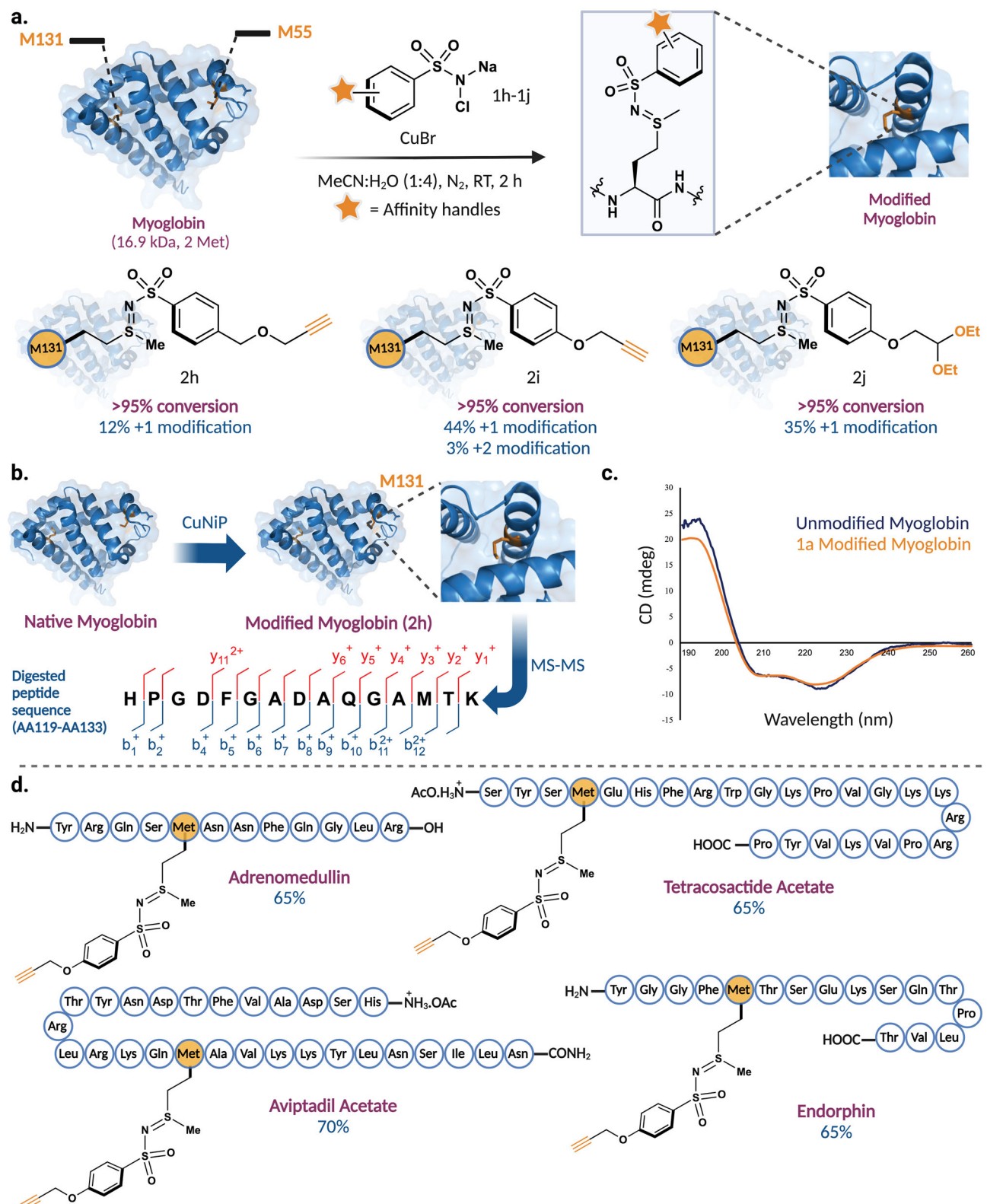

**Fig. 6 | Application of CuNiP for selective functionalization of myoglobin and bioactive peptides by affinity tags. a** Installation of various affinity tags on myoglobin by reaction with chloramine-T affinity probes, **1h-1j** to generate sulfonyl sulfimide conjugates **2h-2j**. **b** MS/MS analysis of **1h**-modified myoglobin, **2h** showed modification at M131. **c** Circular Dichroism (CD) analysis of **1a**-modified myoglobin showed that the tertiary structure remained intact as compared to unmodified myoglobin. **d** Substrate scope of CuNiP in selective modification of methionine in bioactive peptides with an affinity tag **1i**-probe. Figure 6, created with BioRender.com, released under a Creative Commons Attribution-NonCommercial-NoDerivs 4.0 International license" (Agreement number: HX26NYAPQ4).

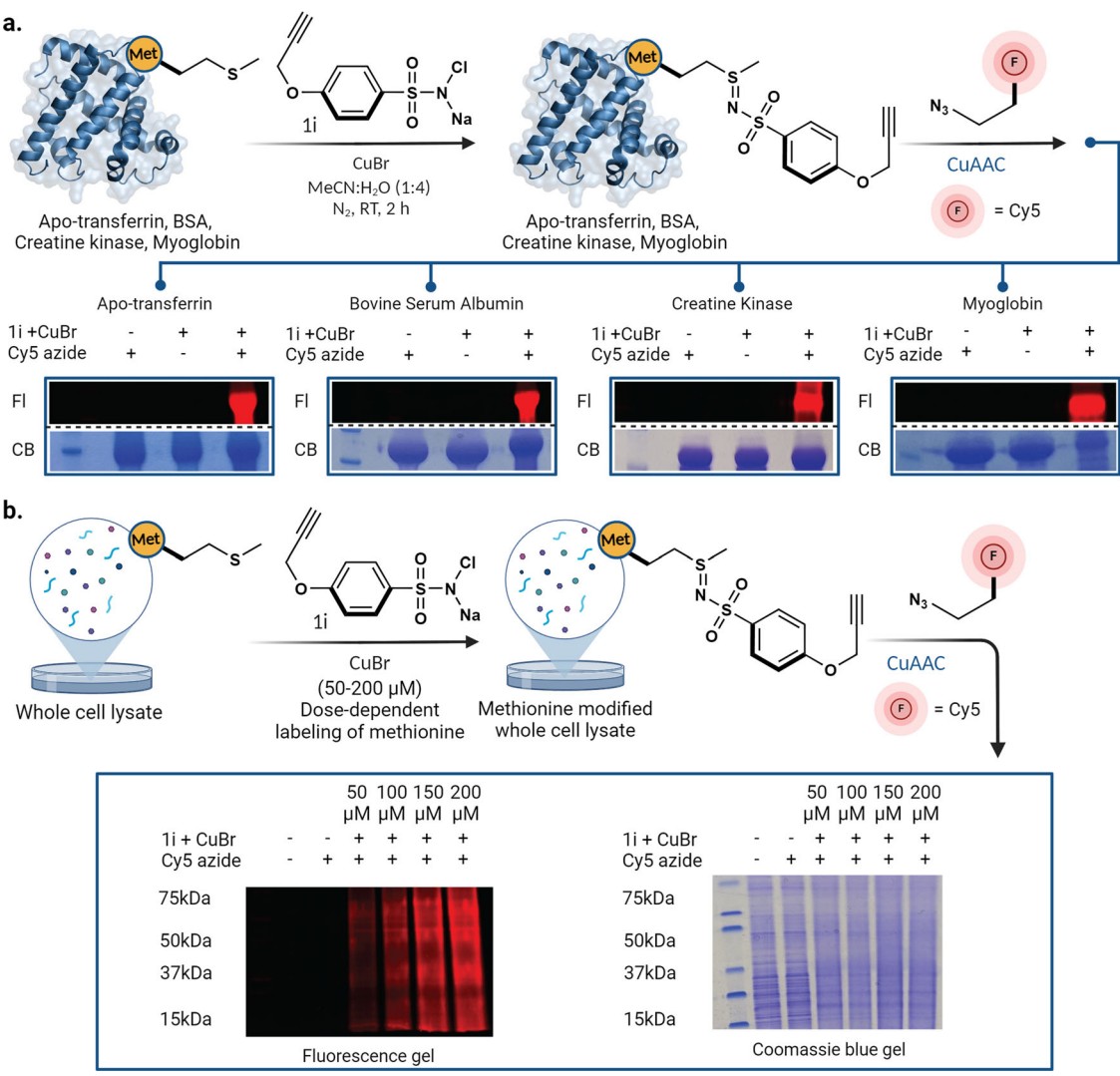

**Fig. 7 | Labeling of native proteins and cell lysate by fluorophore using CuNiP.**
**a** labeling of proteins with alkyne probe **1i** using CuNiP followed by click reaction with azide fluorophore and analysis by in-gel fluorescence. **b** Dose-dependent labeling of a complex cell lysate with **1i** using CuNiP followed by click reaction with azide fluorophore as analyzed by in-gel fluorescence. This experiment was repeated (*n* = 2 biological replicates) with similar results. CB coomassie blue, Fl fluorescence gel Source data are provided as a Source Data file. Figure 7, created with BioRender.com, released under a Creative Commons Attribution-NonCommercial-NoDerivs 4.0 International license" (Agreement number: ZB26NYAH42).

acid metabolism proteins (Supplementary Fig. 35). This result suggests a plausible protective role of methionine in gene regulation and cell division. These observations further confirm the ability of CuNiP to profile oxidation-sensitive methionine residues within the human proteome. Consequently, these applications of CuNiP offer a transformative roadmap for unlocking therapeutic avenues and deepening our grasp on the details of methionine biochemistry.

**Live cell labeling and confocal imaging of CuNiP-modified cells**
Leveraging the capabilities of the CuNiP, we utilized it for labeling methionine residues in their native states within live cells - a critical milestone that has the potential to understand cellular functionalities. Initial efforts focused on optimizing and evaluating the cytotoxic effects of CuNiP components in live cells. T-47D cells were exposed to varying concentrations of probe **1i** (100 μM to 2 mM), CuBr (100 μM to 2 mM), and acetonitrile (1%) for a duration of 2 hours, registering a maximum cell death rate of 12.2% at 2 mM concentration (Fig. 9a, Supplementary Fig. 36). The lysis of cells and subsequent reaction with the Cy5 azide fluorophore affirmed dose-dependent labeling of methionine (Fig. 9b, Supplementary Fig. 37). Proteomics analysis of live-cell generated lysate clearly showed a dose-dependent labeling of

methionine residues on proteins (100 μM = 20, 250 μM = 42, 500 μM = 229, 1 mM = 305, 2 mM = 236). We attribute the lower number of proteins observed for 2 mM to be associated with increasing cell death as the concentration of CuNiP reagent increases. Gene ontology (GO) analysis of modified proteins further corroborates the presence of significant membrane modification in live cell samples (Fig. 9c). Furthermore, functional categorization of modified proteins showed a broad diversity in the classes of proteins being modified (Fig. 9d). This result highlights the application of CuNiP reaction for live cell activity-based profiling of methionine sites in proteins. Taken together, these results corroborate the potential application of CuNiP reaction for probing the role of methionine-mediated membrane-bound interactomes, in addition to multimeric protein complexes in their native cellular environment. To gain insights into the spatio-temporal localizations of modified proteins, we embarked on confocal imaging of CuNiP-modified T-47D cells that are labeled with Cy5 azide fluorophore. The results unveiled a dose-dependent increase in fluorescence intensity in modified cells (>50 cells), accentuating pronounced labeling across cell membrane, cytoplasmic, and nuclear regions (Fig. 9e, Supplementary Fig. 38). This not only authenticated CuNiP's capacity for labeling methionines in their native states but also

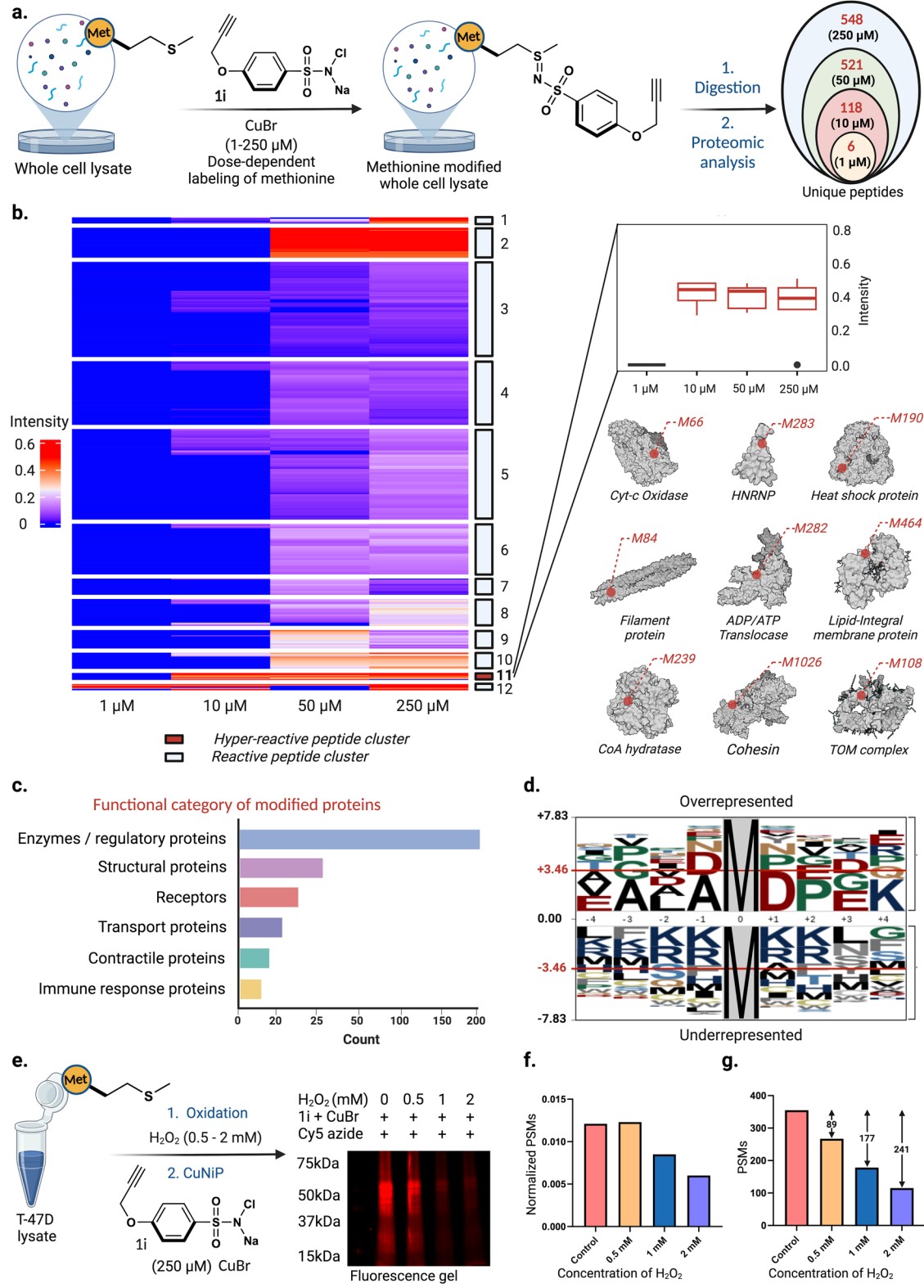

**c.** Functional category of modified proteins

**d.** Overrepresented / Underrepresented

**e.**

**f.**

**g.**

accentuated the potential to unveil previously obscured nuances of cellular dynamics and interplays, shedding light on the intricate spatio-temporal composition of cellular proteins.

In conclusion, we have developed a copper(I)-catalyzed nitrene platform (CuNiP) using the precise reactivity of nitrene to selectively label redox-sensitive methionine residues via sulfonyl sulfimidation. Computational insights showcased that the presence of sulfonyl link-age increased the electron density on sulfimide nitrogen, thereby

increasing the hydrolytic stability of the conjugate \significantly. This innovation is supported by the remarkable ability of CuNiP to tag a broad array of bioactive peptides and intact proteins without altering their intricate three-dimensional structures. Its application extends to complex cell environments, where it unveils previously undiscovered ligandable sites with hyperreactive methionine residues. Sequence analysis of CuNiP-labeled methionines showed a noticeable over-representation of negatively charged residues surrounding

**Fig. 8 | Identification of hyperreactive methionines through chemoproteomic profiling. a** Identification of hyperreactive methionine sites in human proteome using dose-dependent labeling (1-250 μM) with **1i** probe. Excel sheet of analysis is included as Supplementary Data 2. **b** Heatmap visualization of **1i** modified peptides to identify proteins with hyperreactive methionine residues. Cluster 11 was observed to possess 9 proteins with hyperreactive methionine sites with most of the modified proteins possessing enzymatic, regulatory, and structural functions. Centre line – median; box limits contain 50% of data; upper and lower quartiles, 75% and 25%; maximum – greatest value excluding outliers; minimum – least value excluding outliers; outliers – more than 1.5 times of the upper and lower quartiles, with $n = 1$ biological independent sample. **c** Functional category of the identified proteins shows a diverse class of proteins bearing hyperreactive methionines (10 μM) with a maximum labeling of enzymes/regulatory proteins. **d** Sequence analysis of the identified peptides (10 μM dose) demonstrate an overrepresentation of negatively charged residues near the labeled Met and underrepresentation of positively charged and aromatic residues showcasing that CuNiP targets oxidation-prone Met residues. **e** In-gel fluorescence analysis and profiling of oxidation-sensitive methionine residues. A dose-dependent labeling of methionine was observed with increasing $H_2O_2$ concentration, with the highest fluorescence intensity observed in control sample without $H_2O_2$ treatment. **f** Chemoproteomic profiling of oxidation sensitive methionine residues through treatment of T-47D cell lysate samples with $H_2O_2$, followed by subsequent treatment with probe **1i** led to a dose-dependent decrease in normalized PSMs of **1i** labeled peptides with increased $H_2O_2$ concentration. **g** dose-dependent decrease in peptide spectrum matches (PSMs) and intensity were observed as $H_2O_2$ concentration increases (0.5 mM of $H_2O_2$ 88 PSMs <control; 1 mM of $H_2O_2$ 177 PSMs <control; 0.5 mM of $H_2O_2$ 240 PSMs <control). The data utilized for analysis of figure (**e, f, g**) were acquired with $n = 1$ biological independent sample. Excel sheet of analysis is included as Supplementary Data 3. Source data are provided as a Source Data file. Figure 8, created with BioRender.com, released under a Creative Commons Attribution-NonCommercial-NoDerivs 4.0 International license" (Agreement number: BT26NXVCTX).

hyperreactive methionines, a phenomenon that accentuates the vital role of the immediate microenvironment in influencing methionine reactivity. We also demonstrated the application of CuNiP reaction for profiling of oxidation-sensitive methionine residues, thus providing a chemical platform for understanding the protective role of methionine residues in reactive oxygen species (ROS) associated diseases. Furthermore, the CuNiP enabled the labeling of methionine in their native states inside live cells, thus signaling a transformative potential in the therapeutic landscape. These insights, facilitated by chemoproteomic profiling, underscore the indispensable role of CuNiP in identifying molecular targets and pathways, thereby exhibiting a potential to advance our understanding of cellular processes and disease pathogenesis.

## Methods
### Cell culture
Human Breast cancer cell line (T-47D; catalog number ATCC HTB-133) was purchased from ATCC. Authentication was done by ATCC through Tandem Repeats (STR) profiling. Cells were maintained at 37 °C and 5% $CO_2$. T-47D cells were cultured in RPMI supplemented with 10% (V/V) fetal bovine serum (FBS) and 1% (V/V) penicillin/streptomycin (100 μg/mL).

### General procedure for sulfonyl sulfimidation on methionine using Ph-I = N-Ts/Cu-salt system
Fmoc-MF-OMe (2 mg, 3.7 μmol, 1.0 equiv), Ph-I = N-Ts (13.8 mg, 37 μmol, 10.0 equiv), and Cu-salt (0.74 μmol, 0.2 equiv) were dissolved in either MeCN (400 μL) or MeCN:H₂O (1:1, 400 μL) and stirred at RT for 3 h. The crude reaction mixture was analyzed using HPLC (Gradient: 0-80% solvent B over 30 min, solvent B: 0.1% formic acid in MeCN) and MS.

### General procedure for chemoselectivity on FXV-CONH₂ (X=reactive amino acids)
FXV-CONH₂ (X = Y, N, C, R, W, K, H, D, S) (3 μmol, 1.0 equiv), Ph-I = N-Ts (11 mg, 30 μmol, 10.0 equiv), and CuI (0.115 mg, 0.6 μmol, 0.2 equiv) were dissolved in MeCN:H₂O (1:1, 400 μL) and stirred at RT for 2 h. The crude reaction mixture was analyzed using HPLC (Gradient: 0-80% solvent B over 30 min, solvent B: 0.1% formic acid in MeCN) and MS.

### General procedure for sulfonyl sulfimidation on methionine using 1a
Fmoc-VKQMK-CONH₂ (1 mg, 1.17 μmol, 1.0 equiv) and **1a** (0.53 mg, 2.3 μmol, 2.0 equiv) were dissolved in MeCN:H₂O (1:1, 200 μL) and stirred at RT for 2 h. The crude reaction mixture was analyzed on HPLC (Gradient: 0-70% solvent B over 30 min, solvent B: 0.1% formic acid in MeCN) and MS.

### General procedure for chemoselectivity studies under ionic mechanism
F**X**V-CONH₂ (X = Y, K) (2.3 μmol, 1.0 equiv) and **1a** (1 mg, 4.6 μmol, 2.0 equiv) were dissolved in MeCN:H₂O (4:1, 400 μL) and stirred at RT for 1 h. The crude reaction mixture was analyzed on HPLC (Gradient: 0-80 % solvent B over 30 min, solvent B: 0.1% formic acid in MeCN) and MS.

### General procedure for the screening of halogen scavengers
FKV-CONH₂ (2 mg, 5.0 μmol, 1.0 equiv), halogen scavenger (150 μmol, 30.0 equiv), and **1a** (2.27 mg, 10.0 μmol, 2.0 equiv) were dissolved in MeCN:H₂O (4:1, 400 μL) and stirred at RT for 1 h. The crude reaction mixture was analyzed on HPLC (Gradient: 0-70% solvent B over 30 min, solvent B: 0.1% formic acid in MeCN) and MS.

### General procedure for the screening of metal salts for sulfonyl sulfimidation of methionine
Fmoc-VKQMK-CONH₂ (1 mg, 1.17 μmol, 1.0 equiv), metal salts (5.8 μmol, 5.0 equiv), and **1a** (1.33 mg, 5.8 μmol, 5.0 equiv) were dissolved in MeCN:H₂O (1:1, 200 μL) under nitrogen atmosphere and stirred at RT for 5 h. The reaction was quenched by adding 10 μL of 0.5 M HCl and analyzed on HPLC (Gradient: 0-70% solvent B over 30 min, solvent B: 0.1% formic acid in MeCN) and MS.

### Chemoselectivity studies for sulfonyl sulfimidation of methionine
Fmoc-KQYWCREHS-CONH₂ (1 mg, 0.7 μmol, 1.0 equiv), CuBr (0.5 mg, 3.5 μmol, 5.0 equiv), and **1a** (0.795 mg, 3.5 μmol, 5.0 equiv) were dissolved in MeCN:H₂O (1:4, 500 μL) under nitrogen atmosphere and stirred at RT for 24 h. The reaction was quenched with 10 μL of 0.5 M HCl and analyzed on HPLC (Gradient: 0-70% solvent B over 30 min, solvent B: 0.1% formic acid in MeCN) and MS.

Fmoc-KQYWCRMEHS-CONH₂ (1 mg, 0.6 μmol, 1.0 equiv), CuBr (0.5 mg, 3.0 μmol, 5.0 equiv), and **1a** (0.75 mg, 3.0 μmol, 5.0 equiv) were dissolved in MeCN:H₂O (1:4, 500 μL) under nitrogen atmosphere and stirred at RT for 5 h. The reaction was quenched with 10 μL of 0.5 M HCl and analyzed on HPLC (Gradient: 0-70% solvent B over 30 min, solvent B: 0.1% formic acid in MeCN) and MS.

### General procedure for the modification of proteins under CuNiP platform
Intact proteins (0.12 μmol, 1.0 equiv) were dissolved in MeCN:H₂O (1:4, 800 μL) and CuBr (12 mM in MeCN, 100 μL, 1.2 μmol), **1** (12 mM in H₂O, 100 μL, 1.2 μmol) were added sequentially. The reaction mixture was stirred at 25 °C for 2 h under nitrogen atmosphere followed by the addition of 10 μL of 0.5 N HCl. The crude reaction mixture was passed through Amicon Ultra 3 kDa spin-concentrator and washed with H₂O (7 × 0.5 mL) to remove the small molecule impurities. Labeled proteins were lyophilized, redissolved in 0.1% formic acid in H₂O, and analyzed using LC-MS.

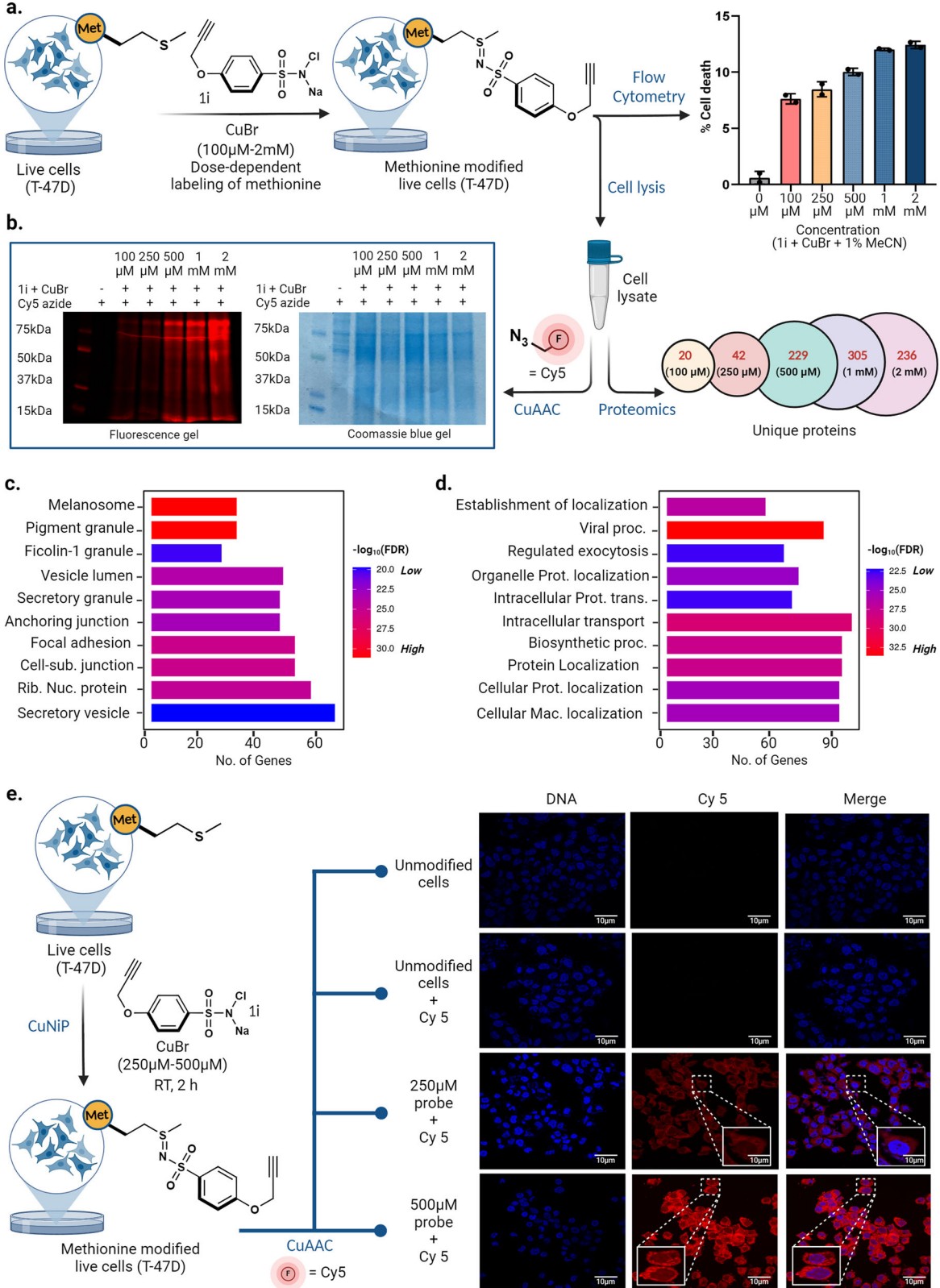

## General procedure for the installation of various payloads onto Myoglobin

Myoglobin (2 mg, 0.12 μmol, 1.0 equiv) was dissolved in MeCN:H$_2$O (1:4, 800 μL) and CuBr (12 mM in MeCN, 100 μL, 1.2 μmol), **1h-1j** (12 mM in H$_2$O, 100 μL, 1.2 μmol) were added sequentially. The reaction mixture was incubated at 25 °C for 2 h under nitrogen atmosphere followed by the addition of 10 μL of 0.5 N HCl. The crude reaction mixture was passed through Amicon Ultra 3 kDa spin-concentrator and washed with H$_2$O (7×0.5 mL) to remove the small molecule impurities. This labeled protein was lyophilized, redissolved in 0.1% formic acid in H$_2$O, and analyzed using LC-MS.

**Fig. 9 | Dose dependent methionine labeling in live cells: cytotoxicity analysis and protein localization insights. a** Dose dependent labeling in live T-47D cells and cytotoxicity analysis showed less than 15% cell death even with 2 mM concentration of CuNiP reagents. Data are represented as mean ± SD (n = 2 biological replicates). **b** Lysis, and in-gel fluorescence analysis of T-47D cell lysates obtained by labeling proteins in live T-47D cells using CuNiP clearly showed a dose-dependent modification of proteins. Proteomics analysis of modified lysates also demonstrated a dose-dependent labeling of proteins with **1i** (100 μM = 20 proteins; 250 μM = 42 proteins; 500 μM = 229 proteins; 1 mM = 305 proteins; 2 mM = 236 proteins). This experiment was repeated (*n* = 3 biological replicates) with similar results. Excel sheet of analysis is included as Supplementary Data 4. **c** Gene ontology analysis of cellular compartment of modified proteins showed a significant modification of membrane-bound proteins, in addition to cytoplasmic proteins within the cell. **d** Gene ontology analysis of biological processes of modified proteins showed a significant modification of proteins of diverse biological functions. **e** Confocal imaging of CuNiP-modified T-47D cells (>50 cells) showed the dose-dependent modification of proteins inside cells with a wide spatio-temporal distribution (membrane, cytoplasm, and nuclear proteins). This experiment was repeated (n = 2 biological replicates) with similar results. Source data are provided as a Source Data file. Figure 9, created with BioRender.com, released under a Creative Commons Attribution-NonCommercial-NoDerivs 4.0 International license" (Agreement number: YB26NY9L7L).

## General procedure for CuNiP modification of proteins with azide fluorophore

To 120 μM stock solution of proteins in MeCN:H$_2$O (1:4, 400 μL) was added 1.2 mM of probe **1i** and CuBr. The reaction was stirred at room temperature for 2 h. Samples were filtered using a 3 kDa molecular weight cut-off filter to obtain pure proteins. **1i** labeled proteins were dissolved in 100 μL of water, followed by the addition of 50 μL of 100 mM TBTA in water, 50 μL of 100 mM freshly prepared ascorbic acid in the water, 50 μL of 50 mM of CuSO$_4$ in water, and 2 μL of 10 mM Cy5 azide in DMSO. The reaction was stirred for 1 h and filtered with a 3 kDa filter, followed by analysis of proteins through in-gel fluorescence imaging and Coomassie blue staining. Samples were loaded on a Novex WedgeWell 4–20% Tris-Glycine gel. Gel was run in Tris-glycine running buffer at 180 V. The gel was then stained with Coomassie brilliant blue for 1 h and destained overnight.

## General procedure for dose-dependent CuNiP modification of lysates and conjugation with Cy5 azide fluorophore

To 4 tubes (individual reactions) of 100 μg of lysate in degassed MeCN:H$_2$O (1:4, 400 μL) were treated with freshly prepared 50 μM, 100 μM, 150 μM, and 200 μM CuBr that had been re-suspended in 50 μL of acetonitrile. To the 4 samples were added freshly prepared 50 μM, 100 μM, 150 μM, and 200 μM probe **1i** that had been re-suspended in 100 μL of NaP buffer pH 7. The reaction was stirred at room temperature for 3 h. Upon completion of reaction, samples were acetone precipitated, followed by Cy5 azide fluorophore labeling. Proteins were dissolved in 100 μL of water, followed by the addition of 50 μL of 100 mM TBTA in water, 50 μL of freshly prepared 100 mM ascorbic acid in water, 50 μL of 50 mM of CuSO$_4$ in water, and 2 μL of 10 mM Cy5 azide in DMSO. The reaction was stirred for 1 h and acetone precipitated, followed by analysis of proteins through in-gel fluorescence imaging and Coomassie blue staining. Samples were loaded on a Novex WedgeWell 4–20% Tris-Glycine gel. Gel was run in Tris-glycine running buffer at 180 V. The gel was then stained with Coomassie brilliant blue for 1 h and destained overnight.

## General procedure for dose-dependent CuNiP modification of lysates and proteomics analysis

To 4 tubes (individual reactions) of 100 μg of lysate in 400 μL of MeCN:H$_2$O (1:4) were treated with freshly prepared solution of CuBr and **1i** (1 μM for super low dose, 10 μM for low dose, 50 μM for medium dose, and 250 μM for high dose). The reaction was stirred at room temperature for 2 h. The proteins were acetone precipitated, followed by digestion using SMART Digest™ Trypsin Kit by Thermo Scientific. Digested samples were resuspended in Buffer A (0.1% FA in water), and the peptide amount was determined by Pierce™ Quantitative Peptide Assays & Standards (Thermo Fisher Scientific) according to manufacturer instructions. Samples were injected into a nanoElute UPLC autosampler (Bruker Daltonics) coupled to a timsTOF Pro2 mass-spectrometer (Bruker Daltonics). The peptides were loaded on a 25 cm Aurora ultimate CSI C18 column (IonOpticks) and chromatographic separation was achieved using a linear gradient starting with a flow rate of 250 nL/min from 2% Buffer B (0.1% FA in MeCN) and increasing to 13% in 42 min, followed by an increase to 23% B in 65 min, 30% B in 70 min, then the flow rate was increased to 300 nL/min and 80% B in 85 min, this was kept for 5 min. The mass-spectrometer operated in positive polarity for data collection using a data-dependent acquisition (ddaPASEF) mode. The cycle time was 1.17 s and consisted of one full scan followed by 10 PASEF/MSMS scans. Precursors with the intensity of over 2500 (arbitrary units) were picked for fragmentation, and precursors over the target value of 20,000 were dynamically excluded for 1 min. Precursors below 700 Da were isolated with a 2 Th window and ones above with 3 Th. All spectra were acquired within an m/z range of 100 to 1700, and fragmentation energy was set to 20 eV at 0.6 1/K0 and 59 eV at 1.60 1/K0.

## Flow cytometry analysis of cell death by Probe 1i, CuBr, and 1% MeCN

T-47D cells were treated with probe **1i** (100 μM to 2 mM), CuBr (100 μM to 2 mM), and 1% acetonitrile for 2 h. After incubation, cells were washed with PBS, detached with trypsin, and stained with Annexin V/PI, according to the manufacturer's protocol. Annexin V (AV) conjugated to FITC was used to determine apoptosis, and propidium iodide (PI) was used to determine necrosis within the cell population. Cells were analyzed via flow cytometry within 1 h to quantify cell death. FlowJo software (version 10.8.1) was used to analyze data collected on the cytometer. PI and AV controls were used to determine quadrant placement. All the experiments were performed in triplicate.

## Dose-dependent fluorophore labeling and proteomic analysis of live cells

Live T-47D cells were plated on 6 cm petri dishes supplemented with RPMI media and incubated for 24 h. Cells were then treated with probe **1i** (100 μM to 2 mM), CuBr (100 μM to 2 mM), and 1% acetonitrile for 2 h. After 2 h, cells were washed 3 times with cold PBS and lysed using RIPA buffer (50 mM Tris HCl [pH 8], 150 mM NaCl, 1% NP-40, 0.5% sodium deoxycholate, 0.1% SDS) supplemented with protease and phosphatase inhibitors. Lysates were centrifuged 6500x *g*, 10 min at 4 °C, and soluble lysate was collected. To 100 μg of lysate in 100 μL of PBS buffer were treated with 50 μL of 100 mM TBTA in water, 50 μL of freshly prepared 100 mM ascorbic acid in water, 50 μL of 50 mM of CuSO$_4$ in water, and 2 μL of 10 mM Cy5 azide in DMSO. The reaction was stirred for 1 h and acetone precipitated, followed by analysis of proteins through in-gel fluorescence imaging and Coomassie blue staining. Samples were loaded on a Novex WedgeWell 4–20% Tris-Glycine gel. Gel was run in Tris-glycine running buffer at 180 V. The gel was then stained with Coomassie brilliant blue for 1 h and destained overnight. For proteomics analysis, 100 μg of live-cell derived lysates were digested using SMART Digest™ Trypsin Kit by Thermo Scientific. Proteomics analysis was done according to the general protocol described above.

## Confocal Microscopy Imaging of CuNiP labeled T47D cells

Live T-47D cells were plated on microscope slides in a 6 cm petri dish supplemented with RPMI media and incubated for 24 h. Cells were then treated with probe **1i** (100 μM to 2 mM), CuBr (100 μM to 2 mM), and 1% acetonitrile for 2 h. After 2 h, cells were washed 3 times with cold PBS and fixed in 4% formaldehyde solution for 10 min. Cells were subsequently washed 3 times with PBS (5 min) and permeabilized with freshly prepared 0.1% Triton-X solution in PBS. Alkyne-labeled proteins within cells were labeled with Cy5 azide fluorophore using click chemistry. For labeling through click chemistry, fixed and permeabilized cells were incubated in 5 mL of PBS followed by the addition of 50 μL of 100 mM TBTA in water, 50 μL of freshly prepared 100 mM ascorbic acid in water, 50 μL of 50 mM of $CuSO_4$ in water, and 1 μL of 10 mM Cy5 azide in DMSO. The reaction was stirred for 1 h and washed 3 times with PBS. Nuclear staining of cells was done with Fluoroshield-DAPI mounting media and subsequently imaged on a Leica SP8 confocal microscope. The images were processed and analyzed using ImageJ software to determine the relative labeling of cells.

## LC-MS/MS

Digested samples were resuspended in Buffer A (0.1% FA in water), and the peptide amount was determined by Pierce™ Quantitative Peptide Assays & Standards (Thermo Fisher Scientific) according to manufactures instructions. Samples were injected into a nanoElute UPLC autosampler (Bruker Daltonics) coupled to a timsTOF Pro2 mass-spectrometer (Bruker Daltonics). The peptides were loaded on a 25 cm Aurora ultimate CSI C18 column (IonOpticks) and chromatographic separation was achieved using a linear gradient starting with a flow rate of 250 nL/min from 2% Buffer B (0.1% FA in MeCN) and increasing to 13% in 42 min, followed by an increase to 23% B in 65 min, 30% B in 70 min, then the flow rate was increased to 300 nL/min and 80% B in 85 min, this was kept for 5 min. The mass-spectrometer operated in positive polarity for data collection using a data-dependent acquisition (ddaPASEF) mode. The cycle time was 1.17 s and consisted of one full scan followed by 10 PASEF/MSMS scans. Precursors with intensity of over 2500 (arbitrary units) were picked for fragmentation and precursors over the target value of 20,000 were dynamically excluded for 1 min. Precursors below 700 Da were isolated with a 2 Th window and ones above with 3 Th. All spectra were acquired within an m/z range of 100 to 1700 and fragmentation energy was set to 20 eV at 0.6 1/K0 and 59 eV at 1.60 1/K0.

## Database search (MSFragger)

MS raw files were searched FragPipe GUI version 20 with MSFragger (version 3.8) as the search algorithm. Protein identification was performed with the human Swisspot database (20'456 entries) with acetylation (N-terminus), and oxidation on methionine was set variable modification. To account for the mass shift introduced by the different chemical handles a variable mass shift of 209.0147 Da or 169.0197 Da on Methionine with a maximal occurrence of 3 for the Alkyne **1i** or Methyl **1a**, respectively, was set. Carbamidomethylation of cysteine residues was considered a fixed modification. Trypsin was set as the enzyme with up to two missed cleavages. The peptide length was set to 7–50, and the peptide mass range of 500–5000 Da. For MS2-based experiments, the precursor tolerance was set to 20 ppm and fragment tolerance to 20 ppm. Peptide spectrum matches (PSMs) were adjusted to a 1% false discovery rate using Percolator as part of the Philosopher toolkit (v5). For label-free quantification, match-between-runs were enabled. All downstream analysis was performed in R (version 2023.03.0). Individual samples were normalized to the mean of all quantified peptides.

## General computational details

The calculations in this study utilized the Gaussian-16 software package. The geometry of all structures were optimized using the B3LYP-D3(BJ)/[6-31 G(d,p)] level of theory, which combines the B3LYP density functional with Grimme's empirical dispersion-correction (D3) and Becke-Johnson (BJ) damping-correction. The split-valence 6–31 G(d,p) basis sets were employed for all atoms. Frequency analyses were conducted at the same level as the geometry optimization to characterize the minimum structures and to include enthalpy and entropy corrections. The effects of the solvent were considered by incorporating bulk solvent effects using the SMD model, with water chosen as the solvent. The reported thermodynamic data were calculated at a temperature of 298.15 K and a pressure of 1 atm.

## Reporting summary

Further information on research design is available in the Nature Portfolio Reporting Summary linked to this article.

## Data availability

All data supporting the findings of this study are available within the supplementary information. These include the synthesis of the probes, optimization of the reaction condition, the procedure of sulfonyl sulfimidation on peptides, intact proteins, cell lysates, chemoselectivity studies, stability studies, DFT calculations, chemoproteomic profiling, uncropped gel blot scans, and product characterization by NMR, HPLC, LC-MS, MS/MS, and HRMS. The mass spectrometry proteomics data generated in this study have been deposited to the ProteomeXchange Consortium via the PRIDE partner repository with the dataset identifier PXD051224. Protein identification was performed with the human Swisspot database (20'456 entries) [https://www.uniprot.org/uniprotkb?query=*&facets=model_organism%3A9606] Source data are provided as a Source Data file. Supplementary data are provided with this paper. Source data are provided with this paper.

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

## Acknowledgements

This research was supported by NIH (grant No. 1R35GM133719-01) and NSF (Grant No. CHE-2108774) to M.R. Research Scholar Grant, RSG-22-025-01-CDP, from the American Cancer Society supported M.R. Swedish Research Council (grant 2023-00510) supported C.M.B. B.E. acknowledges the use of the resources of the Cherry Emerson Center for Scientific Computation at Emory University.

## Author contributions

S.S., B.E., and M.R conceptualized the work. S.S. designed and performed small molecule synthesis, synthesized various electronically diverse probes, optimized the reaction conditions through ionic and nitrene pathway using chloramine-T, carried out chemoselectivity studies, stability studies, bio-active peptide and protein labeling experiments, affinity tag attachment, each of which was characterized using NMR, HRMS and LCMS and MS/MS. B.E. designed and performed

optimization of the reaction conditions using *N*-tosyliminoiodinane, chemoselectivity studies, stability studies, fluorophore labeling of proteins and cell lysate, cell viability studies, and confocal imaging of labeled cells. B.E. designed, performed, and analyzed DFT calculations, and wrote computational section. S.S and B.E. carried out chemoproteomic analysis on cell lysate and live cells. C.M.B., P.B., and D.E.G. acquired and analyzed proteomics data. All authors analyzed the results. S.S., B.E and M.R. wrote the manuscript. M.R. acquired funding and supervised the project. #S.S and B.E. contributed equally to this work.

## Competing interests

B.E. and M.R. are the inventors of International PCT Application No. PCT/US2022/081837 of Emory University covering the compositions and methods for labeling of Thioethers and its application in global profiling and identification of methionine sites from a complex biological system. There are no other competing interests to declare.
