## [Peer Review File · Nature Communications]

Copper(I)-Nitrene Platform for Chemoproteomic Profiling of MethionineEditorial Note: The schematic figures on pages 32 and 33 were created with BioRender.com, released under a Creative Commons Attribution-NonCommercial-NoDerivs 4.0 International license.

REVIEWER COMMENTS

Reviewer #1 (Remarks to the Author):

This paper develops a copper-mediated nitrene transfer process for methionine modification, producing N-sulfonyl sulfimide products. The reactivity study is enlightening, and the use of simple chloramine compounds may well make for simpler synthesis compared to hypervalent iodine or oxaziridines as precursors to nitrenoids. The method easily allows incorporation of useful chemical handles.

Tools to interrogate subtle differences in reactivity of side chains brought about by local structure are an important fundamental challenge with large implications for chemical biology. The work here adds to the relatively small set of methionine-selective chemistries, and the demonstration of a copper-mediated mechanistic pathway for nitrene transfer to sulfur is significant and could spur significant revision in the way researchers address methionine reactivity challenges.

Although I think this work is a nice achievement that will be useful for the community, I do have some concerns about aspects of the data, the mechanistic discussion, and the lack of a thorough examination of the attributes that separate this method from previous efforts with oxaziridines, which produce quite similar products. I outline questions that should be addressed below:

1. SI p 94-95: The HPLC traces here are quite broad, and the product peak (4.127 min) has nearly identical retention time to the starting material. MS S/N is rather poor, so I'm concerned that quantification is questionable here. Furthermore, what is the unidentified material at retention time ~ 6.5 min?
2. The paper gives insufficient credit to Chang's prior work with nitrene transfer using oxaziridine

reagents. Some important precedent is not cited (although the initial report is cited):

<https://doi.org/10.1021/acscentsci.9b01038> (Ohata... Chang)

<http://dx.doi.org/10.1021/jacs.9b04744> (Christian... Chang)

3. A major motivation for this work is “the poor stability of strained oxaziridines and the resulting sulfimide conjugates towards hydrolysis limits its applications in live cells and for making covalent inhibitors.” In this regard, the stability test (Fig 3) seems insufficient. The studies here were not performed under identical conditions to the Chang work. A direct stability comparison of the N-sulfonyl sulfimides (Chang work) to the present N-aminocarbonyl sulfimides seems necessary, given that the evidence indicates that both are >50% stable after many days.

4. Relative to the Chang work, this effort is plagued by competitive sulfoxidation, lowering yields and producing product mixtures. In that sense, I don't really see how this is a concrete advance on prior art.

5. Care must be taken not to assume too much about the mechanism and role of copper here. Copper nitrene intermediates, although plausible here, are certainly not the only explanation or mechanism. Observation of copper nitrene-like species is extremely limited, and other roles for copper are plausible. Although copper nitrene is a reasonable intermediate to assume, and it certainly provides a motivation to develop the experiments in this direction, computing the structure and HOMO-LUMO gap for a copper nitrene (Fig 3c) feels overly specific given available data.

6. The computed HOMO-LUMO gap of 5.4 eV = 124 kcal/mol: is this a reasonable number for a fast room temp reaction?

7. The way the study on variation in sulfonyl structure (Fig 4) is conducted does not really shed light on structure-reactivity relationships. All reagents give >95% conversion, and fairly subtle variation in selectivity. As it stands, I would either delete Fig 4b, or else re-design the reactivity studies to measure kinetic differences and/or conditions with lower conversions.

8. The mechanistic scheme (Fig 4b) also has concerns. “Hydrolysis” here is unclear and perhaps overly simplistic. Complexation of a thioether group to a copper center should not render the S electrophilic for nucleophilic attack by a water molecule.

9. Copper nitrene transfer chemistry is classically understood as a Cu(I) intermediate (e.g. <https://doi.org/10.1021/ja00126a044>, or <https://doi.org/10.1039/D3SC03641C>, or <https://doi.org/10.1039/10.1002/ijch.201900181>), or sometimes formally as a Cu(III)

species, but the view of a Cu(II) species here is unusual and should be referenced. This point is also relevant to the computed structures as well. Why have the authors assumed Cu(II)? The key intermediates here are often termed carbenoids in the literature because the intermediate is indeed largely unknown.

10. Other issues with the mechanism: the bromide atom disappears in the first arrow, raising questions about conservation of charge if a bromide leaving group is assumed.

11. The mechanism draws specific ligands into some structures and not others, and does not use bracket (e.g. [Cu]) notation to indicate unspecified ligands.

12. The MS data for modified lysozyme, chymotrypsinogen A, creatine kinase, BSA seems to be missing from the SI? How are conversions and yields determined here?

13. Some other protein data is of relatively poor quality. This is common for larger proteins with ensemble mixtures of modifications, but care must be taken in quantification in these cases. What do we make of the peak in apo-transferrin modification at 79906, for example? Is this consistent with a double-modification?

14. How are the ratios of various Met sites measured (Fig 5). Proteomics data of MS peak heights after digestion is not typically quantitative.

15. Some discussion of Fig 8c may be useful. What software was used in its creation? Is it possible that the underrepresentation of K/R residues is an experimental artifact due to changes in trypsin cleavage induced by local modification?

Reviewer #2 (Remarks to the Author):

The authors have developed copper(I)-catalyzed nitrene Platform (CuNiP), which is a highly chemoselective method for methionine labeling to generate highly stable sulfonyl sulfimide conjugates. I'm not familiar with the experimental sections, and thus I only discuss the computational section. They have demonstrated CuNiP selectively labeled methionine residues in a wide range of bioactive peptides, intact proteins and proteins in complex cell lysate mixtures. NBO analysis revealed the origin of the stability of sulfonyl sulfimides, and evaluated the effects of substituents on the chloramine-T in the efficiency of methionine labeling. These findings exhibit a potential to improve the understanding of cellular processes and disease pathogenesis. I recommend this manuscript to be published in this journal after addressing the minor issues.

(1) Page 10, the sentence in lines 255-257 “The results from these analysis showed that the electron density on the nitrogen (QN) of the copper-nitrene complex increases when an electron-donating groups are present (1a; QN-Mulliken = -0.230, 1b; QN-Mulliken = -0.242) hence the increased partial negative charge (Fig. 4b, Supplementary Fig. 18).”. In this passage, the last part “hence the increased partial negative charge” seems incomplete, and thus the authors should reorganize this part.

(2) Page 12, the sentence in lines 310-315. “The ability of CuNiP to selectively modify particular methionine sites among several methionine residues on native proteins This preference for labeling a particular methionine site(s) in the presence of multiple methionine residues on proteins is in line with the chemoproteomics principal where hyperreactive Met undergo selective labeling based on the microenvironment thus shows the remarkable ability of this platform to detect hyperreactive Met in a proteome.” There are serious grammatical errors in this passage, and thus the authors should give a revision.

Reviewer #3 (Remarks to the Author):

In this manuscript, Sadu et al. developed a novel methionine probe based on copper(I)-nitrene and used it to profile breast and prostate cancer cell lines via LC-MS/MS and microscopy. The authors demonstrated a commendable approach in the development of probe 1a, employing a rational design strategy coupled with Density Functional Theory (DFT) calculations. The investigation is very thorough and the probe works as expected on amino acids, peptides, purified proteins, and more complex systems like cell lysates and living cells. However, there are a certain number of aspects that need to be addressed before this manuscript becomes publishable in Nat Commun.

- The experiment shown in Figure 3e should be repeated with the same amino acids on the peptide plus a methionine and in the negative control an alanine instead, as to present a yield for the reactions.
- The authors should calculate the selectivity for methionine from the chemoproteomic experiment presented in Figure 8a by searching for the expected adduct on Met, Lys, Gln, Tyr, Cys, Arg, Asp, His, and Ser and report the result as a bar graph.
- The authors explained in Figure 5 that mostly surface-exposed methionine are labeled. Does this still hold true for the 296 identified Met in the lysate experiment in Figure 8a?

- The concept of hyperreactive methionine is used to describe the probe labeled methionine; however, no data is presented to support this argument. The only explanation explored in Figure 5 was solvent accessibility, which differs from hyperactivity. The authors should consider removing the statement on hyperreactivity or provide data to support it such as calculating intensity ratios between the different tested concentrations from 1 μ M to 250 μ M and the sites harboring a lower or no increase in intensity among concentrations would be hyperreactive and worth discussing in more details. In other words, provide evidence that these residues are saturated at lower concentrations of probe labeling, suggesting hyperreactivity.
- As mentioned other methionine reactive probes exist, notably using an oxaziridine warhead (ReACT platform) and this probe usually identifies around 1000 methionine (He et al., 2022, *Molecular Cell* 82, 3045–3060). Is there any overlap between their identified methionines and the ones identified here? It would add value to this new probe to compare the methionine that can be profiled with these different warheads.
- The fluorescence gel presented in Figure 9b using probe 1i (1a alkyne) for labeling on live cells shows only 2-3 bands around 75 kDa starting at 500 μ M. The authors should perform the LC-MS/MS experiment presented in Figure 8a in live cells and report the number of methionine enriched as currently, it seems that this probe labels very few proteins on live cells which is not the purpose of a broad profile probe.
- How many cells were quantified in the microscopy presented in Figure 9c? Only 3 cells are shown and not quantified. The authors should quantify at least 20 cells. Further, the authors claim that the probe labels cytoplasmic and nuclear proteins. However, from the single image, it is impossible to differentiate between cell surface labeling and intracellular labeling. A Z-stack image should be acquired to support this claim.
- This manuscript shows a very methodological and rational development of probe 1a but an application of the probe would greatly strengthen the study. The authors end this work by showing that live cells can be labeled. A suitable example could be to profile oxidation-sensitive methionines via the platform shown in Figure 8a.

TITLE: Copper(I)-Nitrene Platform for Chemoproteomic Profiling of Methionine

We want to take this time to sincerely thank the reviewers for their insight into bettering our research. We have addressed the reviewers' concerns in a revised version of our manuscript. Please see our responses to the reviewers below.

Reviewer #1 (Remarks to the Author):

This paper develops a copper-mediated nitrene transfer process for methionine modification, producing N-sulfonyl sulfimide products. The reactivity study is enlightening, and the use of simple chloramine compounds may well make for simpler synthesis compared to hypervalent iodine or oxaziridines as precursors to nitrenoids. The method easily allows incorporation of useful chemical handles.

Tools to interrogate subtle different in reactivity of side chains brought about by local structure are an important fundamental challenge with large implications for chemical biology. The work here adds to the relatively small set of methionine-selective chemistries, and the demonstration of a copper-mediated mechanistic pathway for nitrene transfer to sulfur is significant and could spur significant revision in the way researchers address methionine reactivity challenges.

Although I think this work is a nice achievement that will be useful for the community, I do have some concerns about aspects of the data, the mechanistic discussion, and the lack of a thorough examination of the attributes that separate this method from previous efforts with oxaziridines, which produce quite similar products. I outline questions that should be addressed below:

We want to thank the reviewer for these comments and the opportunity to address his/her concerns. We thank them for recognizing the impact of our research.

Comment: SI p 94-95: The HPLC traces here are quite broad, and the product peak (4.127 min) has nearly identical retention time to the starting material. MS S/N is rather poor, so I'm concerned that quantification is questionable here. Furthermore, what is the unidentified material at retention time ~ 6.5 min?

Justification:

We have re-performed the experiment with tetracosactide acetate and alkyne probe **1i**. 2 mg of tetracosactide acetate (final conc. 1.25 mM) was dissolved in MeCN:H₂O and to it were added CuBr (110 μ L from 20 mM stock solution, final conc. 6.25 mM, 5.0 equiv), and **1i** (340 μ L from 10 mM stock solution, final conc. 6.25 mM, 5.0 equiv). The resulting solution was stirred at RT for 3 h and quenched by adding 10 μ L 0.5 N HCl. The crude mixture was analysed using HPLC (2-80% solvent B over 30 min, 1 mL/min, solvent B: 0.1% formic acid in MeCN) to determine the conversion and analyzed by MS. This data has been added into the revised supporting information (Supplementary Fig. **30**).

Unmodified Tetracosactide Acetate NH₂-SYSMEHFRWGKPVGKKRRPVKVYP-OH:
 LCMS *m/z* 1466.7954 (calc. 1466.7966 [M+2H⁺]²⁺), *m/z* 978.1572 (calc. [M+3H⁺]³⁺ = 978.2008), *m/z* 733.9035 (calc. [M+4H⁺]⁴⁺ = 733.9024), *m/z* 587.3240 (calc. [M+5H⁺]⁵⁺ = 587.3234) *m/z* 489.6048 (calc. [M+6H⁺]⁶⁺ = 489.6050) Purity: > 99 % (HPLC analysis at 220 nm). Retention time in HPLC: 8.247 min.

1i modified Tetracosactide Acetate NH₂-SYSMEHFRWGKPVGKKRRPVKVYP-OH:
 LCMS *m/z* 1570.8037 (calc. [M+2H⁺]²⁺ = 1571.3049), 1047.8753 (calc. [M+3H⁺]³⁺ = 1048.8723), *m/z* 786.1547 (calc. [M+4H⁺]⁴⁺ = 786.1553), *m/z* 629.1276 (calc. [M+5H⁺]⁵⁺ = 629.1263) Purity: > 99 % (HPLC analysis at 220 nm). Retention time in HPLC: 6.995 min.

HPLC trace of the unmodified tetracosactide acetate

MS Spectra of unmodified tetracosactide acetate

HPLC trace of the **1i** modified tetracosactide acetate

Signal 1: DAD1 A, Sig=220,4 Ref=off

Peak #	RetTime [min]	Type	Width [min]	Area [mAU*s]	Height [mAU]	Area %
1	6.995	MM	0.8261	9.97348e4	2012.19080	64.1673
2	8.925	MM	0.5254	6.60278e4	2094.67554	34.8327

(Note: The peak at 4.08 min does not correspond to any peptide fragment, therefore we presume that it is coming from organic impurity)

MS spectra of **1i** modified tetracosactide acetate

MS spectra of unidentified impurity at 4.08 min showing no peptide fragment is present.

Comment: The paper give insufficient credit to Chang's prior with nitrene transfer using oxaziridine reagents. Some important precedent is not cited (although the initial report is cited): <https://doi.org/10.1021/acscentsci.9b01038> (Ohata...Chang) <http://dx.doi.org/10.1021/jacs.9b04744> (Christian...Chang)

Justification: Necessary citation has been added in the revised manuscript. (Ref 16 & 17 in the revised manuscript)

Comment: A major motivation for this work is “the poor stability of strained oxaziridines and the resulting sulfimide conjugates towards hydrolysis limits its applications in live cells and for making covalent inhibitors.” In this regard, the stability test (Fig 3) seems insufficient. The studies here were not performed under identical conditions to the Chang work. A direct stability comparison of the *N*-sulfonyl sulfimides (Chang work) to the present *N*-aminocarbonyl sulfimides seems necessary, given that the evidence indicates that both are >50% stable after many days.

Justification:

The primary objective of our work is to develop new, mechanistically and structurally different labelling strategies for methionine and apply them for profiling of hypereactive methionine that cannot be captured by current methionine selective probes. In response to your feedback, we have revised the manuscript to emphasize the novel contributions of our Copper(I)-Nitrene Platform (CuNiP) and its utility in chemoproteomic profiling in both vitro and live cells. Our new studies which are highlighted in details below led to the discovery of new protein labeled sites, identification of novel labelling sites and live cells proteomics. We removed the stability comparison with oxaziridine-based sulfimides, focusing instead on the innovative aspects of our work.

Comment: Relative to the Chang work, this effort is plagued by competitive sulfoxidation, lowering yields and producing product mixtures. In that sense, I don't really see how this is a concrete advance on prior art.

Justification:

We acknowledge the reviewer's concern regarding the competitive sulfoxidation observed in our Copper(I)-Nitrene Platform (CuNiP), particularly in comparison to the work by Chang *et al.* Indeed, the issue of competitive sulfoxidation is a prevalent challenge in the oxidative modification of methionine, as also observed in the oxaziridine probe (Chang *et.al.*, Science **2017**, 355, 597-602, Fig. 2; J. Am. Chem. Soc. **2019**, 141, 12657–12662, supporting information, Page S7). The primary objective of our work is to develop new, mechanistically and structurally different labelling strategies for methionine and apply them for profiling of hypereactive methionine. Till date, excellent reactivity and chemoselectivity of nitrenes have not been harnessed for protein bioconjugation and profiling. We wanted to utilize this concept in chemoselective labeling of methionine. In this context, we have made the following advancements:

1. Discovery of New Protein labeled sites: The comparisons of new identified labelled methionine sites between CuNiP and Chang's oxaziridine probe have yielded promising results. At identical probe concentrations (10 μ M-low dose, 50 μ M-medium dose, and 250 μ M-high dose), the CuNiP platform labeled 63-75% new proteins than oxaziridine probe. Although the two cell lines were different (HeLa for oxaziridine and T47D for CuNiP) reports suggest that they share upto 96% proteome similarity (Mol. Cell. Proteomics, **2012**, 1–11). This indicates a significant extension of the chemoproteomic landscape, unveiling new proteins and potential therapeutic targets.

2. Identification of Novel Labeling Sites: *Within the subset of proteins that were labeled by both CuNiP and the oxaziridine platform, our method was able to identify 55-63% new protein sites (for low and medium dose) across low-dose, medium-dose, and high-dose.* This highlights the ability of CuNiP to label new sites within the same proteins.

3. Live cell Proteomics: Further expanding CuNiP, we performed methionine profiling inside live cells, labelling proteins at their native states. Proteomics analysis of live-cells clearly showed a dose-dependent labelling of methionine residues on proteins with **1i** (Fig. **9c**, revised manuscript; 100 μM = 20 proteins; 250 μM = 42 proteins; 500 μM = 229 proteins; 1 mM = 305 proteins; 2 mM = 236 proteins). Gene ontology (GO) analysis of modified proteins further corroborates the presence of significant membrane modification in live cell samples (Fig. **9c**, revised manuscript). Furthermore, functional categorization of modified proteins showed a broad diversity in the classes of proteins modified (Fig. **9d**, revised manuscript). This result highlights the application of CuNiP for profiling of methionine sites in proteins in live cells.

We believe that the Copper(I)-Nitrene Platform marks a significant step forward towards methionine bioconjugation as well as chemoproteomic profiling. It's ability to identify new proteins and labeling sites, combined with the stability of its conjugates, positions it as a valuable tool. Different labelling strategies may be complementary and collectively enrich the toolkit available to researchers. CuNiP adds a valuable dimension by offering alternative mechanisms for methionine labeling. We hope that our revised manuscript now clearly conveys the innovative aspects and the significance of our method in the context of the existing body of work.

Comment: Care must be taken not to assume too much about the mechanism and role of copper here. Copper nitrene intermediates, although plausible here, are certainly not the only explanation or mechanism. Observation of copper nitrene-like species is extremely limited, and other roles for copper are plausible. Although copper nitrene is a reasonable intermediate to assume, and it certainly provides a motivation to develop the experiments in this direction, computing the structure and HOMO-LUMO gap for a copper nitrene (Fig 3c) feels overly specific given available data.

Justification:

The formation of nitrene species in the presence of copper catalysts, particularly in the context of reactions involving Chloramine-T has been well documented in literature in Aziridination, sulfimidation, C-H amination etc. (*J. Org. Chem.* **1997**, *62*, 6512-6518; *Journal of Molecular Catalysis A: Chemical*, **2002**, 85-89; *Tetrahedron Letters*, **1969**, 3301-3304; *Tetrahedron Lett.* **1997**, *38*, 7453; *Org. Biomol. Chem.*, **2005**, *3*, 107-111). Therefore, we believe that the reaction is proceeding via an nitrene/nitrenoid species. Although there is no

direct evidence in observing the Cu-nitrene species, we observed that in presence of Cu-salts, chlorination of K, Y, N-terminus is totally suppressed, thus further supporting the presence of Cu-nitrene species. Therefore, we believe it is plausibly going via nitrene mechanism.

Comment: The computed HOMO-LUMO gap of 5.4 eV = 124 kcal/mol: is this a reasonable number for a fast room temp reaction?

Justification:

We have re-evaluated our HOMO-LUMO calculations, and the observed gap of 5.4 eV is consistent. A similar trend is observed for the computational analysis of oxaziridine probe with methionine which had a HOMO-LUMO gap of 7.4 eV (170 kcal/mol) as reported in (*J. Org. Chem.* **2017**, *82*, 9765–9772). Furthermore, evaluation of the HOMO-LUMO gap of oxaziridine using density functional theory method B3LYP-D3(BJ), 6-311g++(d,p) basis set, as demonstrated in our manuscript, gave a gap of 5.07 eV (116 kcal/mol). Consequently, we would like to state that the HOMO-LUMO gap only addresses relative reactivity based on frontier orbital gap of optimized low energy conformer geometries of methionine and Cu-Nitrene-Ligand complex. Reactivity of methionine with Cu-Nitrene-Ligand complex is far more complex and requires geometry optimization of interacting species, favourable orientation of reacting species in space, binding energy, enthalpic and entropic changes. This can be achieved by calculating the full mechanistic pathway for methionine and other reactive residues, and this is beyond the scope of the current manuscript. Work in this direction is currently going on in our lab.

Comment: The way the study on variation in sulfonyl structure (Fig 4) is conducted does not really shed light on structure-reactivity relationships. All reagents give >95% conversion, and fairly subtle variation in selectivity. As it stands, I would either delete Fig 4b, or else re-design the reactivity studies to measure kinetic differences and/or conditions with lower conversions.

Justification:

In the revised manuscript, we have redesigned the experiment to understand the role of electron-donating vs electron-withdrawing group in the product outcome of sulfonyl sulfimidation. Since it was not clear to identify the trend on a protein level, we have performed the same on a model pentapeptide Fmoc-VKQMK-CONH₂ under our optimized condition.

Pentapeptide Fmoc-VKQMK-CONH₂ (1 mg, final conc. 2 mM) was incubated with CuBr (10 mM, 5.0 equiv) and **1a-1g** (10 mM, 5.0 equiv) at RT for 5 h under N₂. It was quenched with 20 μ L 0.5 N HCl and the product:sulfoxide ratio was analyzed using HPLC and MS. For electron-donating group (-Me **1a**, -OMe **1b**) we observed an increased product:sulfoxide ratio (87:13 for **1a**, 72:28 for **1b**). For electron-withdrawing groups, however we saw a marked decrease in product:sulfoxide ratio (-Cl, **1c**, 66:34; -COOH, **1d**, 38:62; -NO₂, **1e**, 37:63; -di-F, **1f**, 24:76; -CF₃, **1g**, 54:46). We think the increased electron density on nitrogen for **1a** and **1b**, as calculated by Mulliken population and Natural Bond Orbital (NBO) analysis, enables more efficient capture of the methionine-Cu-nitrene sulfonium complex to form the sulfonyl sulfimides. This has been added into revised manuscript (Fig. **4a**) and supporting information (Supplementary Fig. **15**)

Peak #	RetTime [min]	Type	Width [min]	Area [mAU*s]	Height [mAU]	Area %
1	7.815	MM	0.6319	7.41530e4	1955.71399	13.3464
2	9.330	MM	0.6676	7.02642e4	1754.18481	86.6536

HPLC trace of **1a** modified Fmoc-VKQMK-CONH₂

Peak #	RetTime [min]	Type	Width [min]	Area [mAU*s]	Height [mAU]	Area %
1	8.214	MM	0.4030	1.55027e4	641.07983	28.3298
2	9.244	MM	0.5031	3.92195e4	1299.27991	71.6702

HPLC trace of **1b** modified Fmoc-VKQMK-CONH₂

Fmoc-VKQM(mod)K-CONH₂: LCMS m/z 1023.4745 (calc. $[M+H^+] = 1039.4740$), m/z 520.2410 (calc. $[(M+2H^+)/2] = 520.2406$), Purity: > 95 % (HPLC analysis at 220 nm). Retention time in HPLC: 9.244 min for product.

MS of **1b** modified Fmoc-VKQMK-CONH₂

Peak #	RetTime [min]	Type	Width [min]	Area [mAU*s]	Height [mAU]	Area %
1	8.235	MM	0.5200	2.67718e4	858.06958	34.2180
2	9.710	MM	0.5143	5.14671e4	1668.02771	65.7820

HPLC trace of **1c** modified Fmoc-VKQMK-CONH₂

Fmoc-VKQM(mod)K-CONH₂: LCMS m/z 1043.4246 (calc. $[M+H]^+$ = 1043.4244), m/z 522.2157 (calc. $[(M+2H^+)/2]$ = 522.2158), Purity: > 95 % (HPLC analysis at 220 nm). Retention time in HPLC: 9.710 min for product.

MS of **1c** modified Fmoc-VKQMK-CONH₂

Peak #	RetTime [min]	Type	Width [min]	Area [mAU*s]	Height [mAU]	Area %
1	7.804	MM	0.8205	8.61880e4	1750.63684	62.4720
2	8.789	MM	0.4991	5.17747e4	1728.86890	37.5280

HPLC trace of **1d** modified Fmoc-VKQMK-CONH₂

Fmoc-VKQM(mod)K-CONH₂: LCMS *m/z* 1053.4536 (calc. $[M+H]^+$ = 1053.4532), *m/z* 527.2316 (calc. $[(M+2H^+)/2]$ = 527.2302), Purity: > 95 % (HPLC analysis at 220 nm). Retention time in HPLC: 8.789 min for product.

MS of **1d** modified Fmoc-VKQMK-CONH₂

Peak #	RetTime [min]	Type	Width [min]	Area [mAU*s]	Height [mAU]	Area %
1	7.906	MM	0.7607	1.01101e5	2215.04761	63.4309
2	9.480	MM	0.6184	5.82867e4	1570.79346	36.5691

HPLC trace of **1e** modified Fmoc-VKQMK-CONH₂

Fmoc-VKQM(mod)K-CONH₂: LCMS m/z 1054.4479 (calc. $[M+H]^+$ = 1054.4485), m/z 527.7276 (calc. $[(M+2H^+)/2]$ = 527.7279), Purity: > 95 % (HPLC analysis at 220 nm). Retention time in HPLC: 9.480 min for product.

MS of **1e** modified Fmoc-VKQMK-CONH₂

Peak #	RetTime [min]	Type	Width [min]	Area [mAU*s]	Height [mAU]	Area %
1	8.429	MM	0.5327	5.31393e4	1662.51257	75.6938
2	9.930	MM	0.3478	1.70637e4	817.75208	24.3062

HPLC trace of **1f** modified Fmoc-VKQMK-CONH₂

Fmoc-VKQM(mod)K-CONH₂: LCMS *m/z* 1045.4458 (calc. $[M+H]^+$ = 1045.4446), *m/z* 523.2246 (calc. $[(M+2H^+)/2]$ = 523.2259), Purity: > 95 % (HPLC analysis at 220 nm). Retention time in HPLC: 9.930 min for product.

MS of **1f** modified Fmoc-VKQMK-CONH₂

Peak #	RetTime [min]	Type	Width [min]	Area [mAU*s]	Height [mAU]	Area %
1	7.672	MM	0.8243	8.36932e4	1692.25208	45.8301
2	9.745	MM	0.8195	9.89230e4	2011.95703	54.1699

HPLC trace of **1g** modified Fmoc-VKQMK-CONH₂

Fmoc-VKQM(mod)K-CONH₂: LCMS m/z 1077.4519 (calc. $[M+H]^+$ = 1077.4508), m/z 539.2279 (calc. $[(M+2H^+)/2]$ = 539.2290), Purity: > 95 % (HPLC analysis at 220 nm). Retention time in HPLC: 9.745 min for product.

MS spectra of **1g** modified Fmoc-VKQMK-CONH₂

Comment: The mechanistic scheme (Fig 4b) also has concerns. “Hydrolysis” here is unclear and perhaps overly simplistic. Complexation of a thioether group to a copper center should not render the S electrophilic for nucleophilic attack by a water molecule.

Response: We would like to state that the proposed pathway represents a plausible mechanistic explanation (supported with DFT) for the observation of sulfonyl sulfimide and sulfoxide products under CuNiP reaction. To evaluate if the sulfur atom of methionine becomes electrophilic upon reaction with CuNiP probe, we optimized the geometries and calculated the electrostatic potential (ESP) maps of methionine and Met-Cu-Nitrene sulfonium complex. Analysis of the ESP map clearly shows the partial electrophilic nature of the sulfur atom in Met-Cu-Nitrene sulfonium complex (-CH₃ analog, **1a**) as compared to unreacted methionine (Supplementary Fig. 17). Based on this finding, we have modified the pathway to highlight the partial electrophilic character instead of the discrete electrophilic character assigned previously. Furthermore, our experimental result clearly supports the proposed explanation as we observed a high product:sulfoxide ratio for electron-donating groups (-Me, **1a**, 87:13; -OMe **1b** [72:28]) as compared to low product:sulfoxide ratio for electron-withdrawing groups (-Cl, **1c**, 66:34; -COOH, **1d**, 38:62; -NO₂, **1e**, 37:63; -di-F, **1f**, 24:76; -CF₃, **1g**, 54:46).

Comment: Copper nitrene transfer chemistry is classically understood as a Cu(I) intermediate (e.g. <https://doi.org/10.1021/ja00126a044>, or <https://doi.org/10.1039/D3SC03641C>, or <https://doi.org/10.1002/ijch.201900181>), or sometimes formally as a Cu(III) species, but the view of a Cu(II) species here is unusual and should be referenced. This point is also relevant to the computed structures as well. Why have the authors assumed Cu(II)? The key intermediates here are often termed carbenoids in the literature because the intermediate is indeed largely unknown.

Justification: Although mostly Cu-nitrene chemistry has been visualized through Cu(I)-Cu(III) intermediate, the utilization of Cu(II)-salts has also been referenced before (*J. Org. Chem.* **1997**, *62*, 6512-6518; Table 4, entry 3). That's why we screened Cu(II)-salts in Fig. **2b**. Also we have corrected the structures of the Cu-(I) salts in Fig. 3b. It will be (CuOTf)₂·benzene and (CuOTf)₂·toluene in which Cu is in +1 oxidation state.

Comment: Other issues with the mechanism: the bromide atom disappears in the first arrow, raising questions about conservation of charge if a bromide leaving group is assumed.

Justification: We have corrected the mechanism in the revised manuscript (Fig. **4c**). In the first step, reaction of CuBr with **1** generates Cu-(III)-nitrene species with release of HBr and NaCl. This has been added for better clarification.

Comment: The mechanism draws specific ligands into some structures and not others, and does not use bracket (e.g. [Cu]) notation to indicate unspecified ligands.

Justification: We have used brackets ([Cu]) for all the notations in the revised manuscript. Since we are not adding any external ligands, we presume acetonitrile is the co-ordinating ligand in all these cases.

Comment: The MS data for modified lysozyme, chymotrypsinogen A, creatine kinase, BSA seems to be missing from the SI? How are conversions and yields determined here?

Justification: To determine the conversion, we have digested the protein and compared the % quant area from MS/MS for the labelled peptide along with their unlabelled or oxidized fragment of the same sequence.

Comment: Some other protein data is of relatively poor quality. This is common for larger proteins with ensemble mixtures of modifications, but care must be taken in quantification in these cases. What do we make of the peak in apo-transferrin modification at 79906, for example? Is this consistent with a double-modification?

Justification: We have reformed the Apo-transferrin experiment. The peak at 79908 corresponds to [M+O+2mod]. The MS/MS analysis of the digested protein shows labeling of M518 along with M483. The labeling ratio of M483:M518 is 3.3:1.

Apo-transferrin (9.5 mg, 0.12 μmol, 1.0 equiv) was dissolved in MeCN: H₂O (1:4, 800 μL) and CuBr (12 mM in MeCN, 100 μL, 1.2 μmol), **1a** (12 mM in H₂O, 100 μL, 1.2 μmol) were added sequentially. The reaction mixture was incubated at 25 °C for 2 h under nitrogen atmosphere followed by the addition of 10 μL of 0.5 N HCl. The crude reaction mixture was passed through Amicon Ultra 3 kDa spin-concentrator and washed with H₂O (7×0.5 mL) to remove the small molecule impurities. This labeled protein was lyophilized, redissolved in 0.1% formic acid in H₂O and analyzed using LC-MS. Intact mass analysis shows 40% +1 mod, and 25% +2 mod. MS/MS analysis of the digested protein shows labeling ratio of M483:M518 is 3.3:1. This has been added into revised manuscript (Fig. **5**) and supporting information (Supplementary Fig. **27**)

MS spectra of **1a** modified apo-transferrin

Deconvoluted MS spectra of **1a** modified apo-transferrin

MS/MS Analysis of 1a modified apo-transferrin:

Biomolecule 184: LCMGSGGLNLCEPNNK

Biomol	Seq Loc	Rule	Pred Mods	RT	Height	Mass	Tgt Mass	Diff (ppm)
184	A(516-530)	Complete digest, Predicted modifications	Alkylation (iodoacetamide) 2, Met-ChT- 3, Alkylation (iodoacetamide) 10	9.758	199506	1874.7849	1874.7784	3.50

ECC (with sample chromatogram)

Mass Spectrum (with MFE spectrum, if available)

Identified Peptide Sequence: LCMGSGGLNLCEPNNK (AA-516-AA530)

b^+		b^{2+}		AA		y^+		y^{2+}	
114.091340	57.549308	1	L	15					
274.151825	137.579551	2	C+IAA	14	1762.761281	881.884279			
574.212110	287.609693	3	M+mod	13	1602.700797	801.854037			
631.233573	316.120425	4	G	12	1302.640512	651.823894			
718.265602	359.636439	5	S	11	1245.619048	623.313162			
775.287066	388.147171	6	G	10	1158.587020	579.797148			
888.371130	444.689203	7	L	9	1101.565556	551.286416			
1002.414057	501.710667	8	N	8	988.481492	494.744384			
1115.498121	558.252699	9	L	7	874.438565	437.722921			
1275.558606	638.282941	10	C+IAA	6	761.354501	381.180889			
1404.601199	702.804238	11	E	5	601.294016	301.150646			
1501.653963	751.330620	12	P	4	472.251423	236.629350			
1615.696890	808.352083	13	N	3	375.198659	188.102968			
1729.739818	865.373547	14	N	2	261.155732	131.081504			
		15	K	1	147.112804	74.060040			

Comment: How are the ratios of various Met sites measured (Fig 5). Proteomics data of MS peaks heights after digestion is not typically quantitative.

Justification: We have calculated the labeling ratios of various Met sites from calculating % quant area and the ion count ratio of modified peptides (for ref., see *J. Am. Chem. Soc.* **2020**, *142*, 21260–21266, supporting information, page S29).

Comment: Some discussion of Fig 8c may be useful. What software was used in its creation? Is it possible that the underrepresentation of K/R residues is an experimental artifact due to changes in trypsin cleavage induced by local modification?

Response: We appreciate the reviewer's observation. The amino acid sequence logo was generated using the plogo map tool (<https://plogo.uconn.edu/>) [<https://www.nature.com/articles/nmeth.2646>]. We do not attribute the underrepresentation of K/R residues to cleavage specificity, given that trypsin is generally unaffected by post-translational modifications of amino acids, as observed in kinase motif analysis. Furthermore, we only examined peptides with at least 4 residues upstream and downstream of modified methionine sites, thus eliminating the potential artificial underrepresentation of lysine or arginine resulting from trypsin cleavage. However, in very rare cases where a peptide is too short but has a tryptic cleavage, they were not included in the motif, potentially leading to an artificial underrepresentation.

The above explanation has been added in the revised manuscript.

Reviewer #2 (Remarks to the Author):

The authors have developed copper(I)-catalyzed nitrene Platform (CuNiP), which is a highly chemoselective method for methionine labeling to generate highly stable sulfonyl suofimide

conjugates. I'm not familiar with the experimental sections, and thus I only discuss the computational section. They have demonstrated CuNiP selectively labeled methionine residues in a wide range of bioactive peptides, intact proteins and proteins in complex cell lysate mixtures. NBO analysis revealed the origin of the stability of sulfonyl sulfimides, and evaluated the effects of substituents on the chloramine-T in the efficiency of methionine labeling. These findings exhibit a potential to improve the understanding of cellular processes and disease pathogenesis. I recommend this manuscript to be published in this journal after addressing the minor issues.

We want to thank reviewer #2 for all his/her comments and recommendation to publish after minor corrections. We appreciate the opportunity to address the minor concerns raised by reviewer #2, which are as follows.

Comment: Page 10, the sentence in lines 255-257 “The results from these analysis showed that the electron density on the nitrogen (Q_N) of the copper-nitrene complex increases when an electron-donating groups are present (1a; Q_N -Mulliken = -0.230, 1b; Q_N -Mulliken = -0.242) hence the increased partial negative charge (Fig. 4b, Supplementary Fig. 18).” In this passage, the last part “hence the increased partial negative charge” seems incomplete, and thus the authors should reorganize this part.

Justification: We have reorganized the line by using the following- “*The results from this analysis showed that the electron density on the nitrogen (Q_N) of the copper-nitrene complex increases when an electron-donating groups are present (1a; $Q_{N-Mulliken} = -0.230$, 1b; $Q_{N-Mulliken} = -0.242$), therefore increasing the partial negative charge*”. This has been added into the revised manuscript.

Comment: Page 12, the sentence in lines 310-315. “The ability of CuNiP to selectively modify particular methionine sites among several methionine residues on native proteins. This preference for labeling a particular methionine site(s) in the presence of multiple methionine residues on proteins is in line with the chemoproteomics principal where hyperreactive Met undergo selective labeling based on the microenvironment thus shows the remarkable ability of this platform to detect hyperreactive Met in a proteome.” There are serious grammatical errors in this passage, and thus the authors should give a revision.

Justification: We have corrected the grammatical mistake in the revised manuscript by using the following- “*The ability of CuNiP to selectively modify specific methionine sites among several others corroborates with the chemoproteomics principal where hyperreactive methionines undergo selective labeling based on the microenvironment. This shows the remarkable ability of this platform to detect hyperreactive Met in a proteome.*” This has been added into the revised manuscript.

Reviewer #3 (Remarks to the Author):

In this manuscript, Sadu et al. developed a novel methionine probe based on copper(I)-nitrene and used it to profile breast and prostate cancer cell lines via LC-MS/MS and microscopy. The authors demonstrated a commendable approach in the development of probe 1a, employing a rational design strategy coupled with Density Functional Theory (DFT) calculations. The investigation is very thorough and the probe works as expected on amino acids, peptides, purified proteins, and more complex systems like cell lysates and living cells.

However, there are a certain number of aspects that need to be addressed before this manuscript becomes publishable in Nat Commun.

We want to thank reviewer #3 for all his/her comments and recommendation to publish after minor corrections. We appreciate the opportunity to address the minor concerns raised by reviewer #3, which are as follows.

Comment: The experiment shown in Figure 3e should be repeated with the same amino acids on the peptide plus a methionine and in the negative control an alanine instead, as to present a yield for the reactions.

Justification:

In the revised manuscript, we have screened Fmoc-KQYWCR**ME**HS-CONH₂ (with methionine) and Fmoc-KQYWCREHS-CONH₂ (without methionine). Reaction of Fmoc-KQYWCR**ME**HS-CONH₂ with **1a** (5.0 equiv) and CuBr (5.0 equiv), provided 79% sulfimidated product. Whereas Fmoc-KQYWCREHS-CONH₂ (negative control) did not result in formation of any product even after 24 h, demonstrating the chemoselectivity of the CuNiP platform. This has been added in revised Figure 3e and supporting information (Supplementary Fig. 12).

Fmoc-KQYWCRMEHS-CONH₂ (1 mg, 0.6 μ mol, 1.0 equiv), CuBr (0.5 mg, 3.0 μ mol, 5.0 equiv), and **1a** (0.75 mg, 3.0 μ mol, 5.0 equiv) were dissolved in MeCN:H₂O (1:4, 500 μ L) under nitrogen atmosphere and stirred at RT for 5 h. The reaction was quenched with 10 μ L of 0.5 M HCl and analyzed on HPLC (Gradient: 0-70 % solvent B over 30 min, solvent B: 0.1% formic acid in MeCN) and MS. HPLC analysis shows 79% sulfimidated product.

Fmoc-KQYWCRMEHS-CONH₂: LCMS m/z 1588.6826 (calc. $[M+H]^+$ = 1588.6824), m/z 794.8452 (calc. $[(M+2H^+)/2]$ = 794.8448), m/z 530.2335 (calc. $[(M+3H^+)/3]$ = 530.2323). Purity: > 95 % (HPLC analysis at 220 nm). Retention time in HPLC: 11.024 min.

Fmoc-KQYWCRM(mod)EHS-CONH₂: LCMS m/z 1757.7026 (calc. $[M+H]^+$ = 1757.7022), m/z 879.3552 (calc. $[(M+2H^+)/2] = 879.3547$), m/z 586.5734 (calc. $[(M+3H^+)/3] = 586.5722$). Purity: > 95 % (HPLC analysis at 220 nm). Retention time in HPLC: 11.699 min.

HPLC trace of Fmoc-KQYWCRM(mod)EHS-CONH₂

MS Spectra of Fmoc-KQYWCRM(mod)EHS-CONH₂

Peak #	RetTime [min]	Type	Width [min]	Area [mAU*s]	Height [mAU]	Area %
1	10.487	MM	0.1631	1.43402e4	1464.95813	21.2177
2	11.699	MM	0.2020	2.22254e4	1833.51257	78.7823

HPLC trace of **1a** modified Fmoc-KQYWCRMEHS-CONH₂

MS Spectra of **1a** modified Fmoc-KQYWCRMEHS-CONH₂

Comment: The authors should calculate the selectivity for methionine from the chemoproteomic experiment presented in Figure 8a by searching for the expected adduct on Met, Lys, Gln, Tyr, Cys, Arg, Asp, His, and Ser and report the result as a bar graph.

Response: We fully agree with the reviewer's comment. To assess the selectivity for methionine, we performed a database search with the modification on Met, Lys, Gln, Tyr, Cys, Arg, Asp, His, and Ser. A limited number of modified peptides can be identified when searching for other amino acids at varying probe concentrations. We believe these observed modifications are artifacts as we do not observe any reactivity of CuNiP with serine or other reactive residues at peptide and recombinant protein examples. This has been added in revised supporting information (Supplementary Fig. **34**).

Comment: The authors explained in Figure 5 that mostly surface-exposed methionine are labeled. Does this still hold true for the 296 identified Met in the lysate experiment in Figure 8a?

Justification: Regarding solvent accessibility of **1i** modified methionine residues, we have calculated the solvent accessible surface area (SASA) of 208 random peptides from the high dose (250 μM) samples in Fig. 8a. Out of 208 sites tested, 155 sites (74%) were found to be buried whereas 53 sites (26%) were solvent exposed. These results highlight the potential of CuNiP to label methionine residues irrespective of their position on a protein. This has been added to the revised manuscript and supporting information (Supplementary Fig. **34**).

Comment: The concept of hyperreactive methionine is used to describe the probe labeled methionine; however, no data is presented to support this argument. The only explanation explored in Figure 5 was solvent accessibility, which differs from hyperactivity. The authors should consider removing the statement on hyperreactivity or provide data to support it such as calculating intensity ratios between the different tested concentrations from 1 μM to 250 μM and the sites harboring a lower or no increase in intensity among concentrations would be hyperreactive and worth discussing in more details. In other words, provide evidence that these residues are saturated at lower concentrations of probe labeling, suggesting hyperreactivity.

Justification: Based on the valuable comments from the reviewer, we have now incorporated data highlighting hyperreactive peptide clusters for the different concentrations. We have generated a heatmap and peptide cluster map identifying the hyperreactive modified methionine sites, with cluster 11 containing 9 proteins with hyperreactive methionine sites. This has been added in revised manuscript (Fig. **8b**) and supporting information (Supplementary Fig. **34**).

Center line—median; box limits contain 50% of data; upper and lower quartiles, 75 and 25%; maximum—greatest value excluding outliers; minimum—least value excluding outliers; outliers—more than 1.5 times of the upper and lower quartiles

Comment: As mentioned other methionine reactive probes exist, notably using an oxaziridine warhead (ReACT platform) and this probe usually identifies around 1000 methionine (He et al., 2022, Molecular Cell 82, 3045–3060). Is there any overlap between their identified methionines and the ones identified here? It would add value to this new probe to compare the methionine that can be profiled with these different warheads.

Justification: Since the abovementioned report of methionine profiling was carried out on a mouse organoid model, for better comparison, we have compared the protein list labeled by oxaziridine and CuNiP based on the original work by Chang *et al.* (Science 2017, 355, 597-602). At identical probe concentration (10 μ M-low dose, 50 μ M-medium dose, 250 μ M-high dose), CuNiP was able to identify 63-78% new proteins that were not labeled by the oxaziridine platform. Interestingly, within the common protein targets under each probe concentration, CuNiP was able to label 55-63% new protein sites (Low & medium doses). This tells us that, due to its unique labeling mechanism and structurally different warhead, CuNiP was able to identify a substantially large set of proteins and protein sites which was not identified before.

Comment: The fluorescence gel presented in Figure 9b using probe 1i (1a alkyne) for labeling on live cells shows only 2-3 bands around 75 kDa starting at 500 μ M. The authors should perform the LC-MS/MS experiment presented in Figure 8a in live cells and report the number of methionine enriched as currently, it seems that this probe labels very few proteins on live cells which is not the purpose of a broad profile probe.

Response: Looking at the fluorescence gel presented in Figure 9b, we can see why it seems like only 2-3 proteins were modified. We believe that these proteins are abundant proteins as can be seen in the Coomassie gel. However, a close-up look at the fluorescent gel clearly highlights a broad distribution of modified proteins across different molecular weights. Consequently, we also carried out proteomics analysis on the labelled live T47D cells and observed a similar dose dependent labelling (20 proteins- 100 μ M, 62 proteins- 250 μ M, 229 protein- 500 μ M, 305 proteins- 1 mM, 236 proteins- 2 mM). We attribute the lower number of proteins observed for 2 mM to be associated with increasing cell death as concentration of CuNiP reagent increases. Interestingly, GO analysis of these proteins clearly identified a broad range of function and localization of modified proteins within the cell, with a significant number of modified proteins been cell membrane-related proteins. This has been added in revised manuscript (Fig. 9b) and supporting information (Supplementary Fig. 37).

Comment: How many cells were quantified in the microscopy presented in Figure 9c? Only 3 cells are shown and not quantified. The authors should quantify at least 20 cells. Further, the authors claim that the probe labels cytoplasmic and nuclear proteins. However, from the single image, it is impossible to differentiate between cell surface labeling and intracellular labeling. A Z-stack image should be acquired to support this claim.

Response: We appreciate the reviewer's suggestion on quantifying more cells and acquiring a z-stack image highlighting the spatiotemporal localization of modification within cells. Consequently, we have repeated the experiment and quantified >50 cells for control and experimental samples. Also, we have included a z-stack image and a gif supplementary material of median intensity image from 22 slices of CuNiP modified cells. From these recent images, we can clearly observe that most of the modifications were at the cell membrane while few modifications were within the nuclear region of the cells. This observation largely concurs with the proteomics results obtained from live cell experiments. This has been added in revised manuscript (Fig. 9e) and supporting information (Supplementary Fig. 38).

Comment: This manuscript shows a very methodological and rational development of probe 1a but an application of the probe would greatly strengthen the study. The authors end this work by showing that live cells can be labeled. A suitable example could be to profile oxidation-sensitive methionines via the platform shown in Figure 8a.

Justification: We thank the reviewer for suggesting such an exciting application for CuNiP reaction. Consequently, we have successfully probed for oxidation sensitive methionine within the human proteome using CuNiP. This was achieved by treating experimental samples with 0.5-2 mM of hydrogen peroxide for 1 h, followed by labelling with **1i** using CuNiP, in-gel fluorescence analysis, and proteomics analysis. Comparison of experimental samples to control sample (without H₂O₂ treatment) readily shows a decrease in fluorescent intensity, suggesting the ability of CuNiP to modify oxidation sensitive methionine residues. Proteomics analysis further corroborates this observation as we observed a dose-dependent decrease in modified methionine sites as hydrogen peroxide increases (356 peptides-control; 267 peptides-0.5 mM; 178 peptides- 1 mM; 115 peptides- 2 mM) with (0.5 mM of H₂O₂, 88 PSMs < control; 1 mM of H₂O₂, 177 PSMs < control; 0.5 mM of H₂O₂, 240 PSMs < control). Further analysis led to the discovery of 86 sites that were only modified in control samples but not in any of the H₂O₂ concentrations. Interestingly, Gene Ontology analysis of these

proteins containing oxidation sensitive methionine residues (86 sites) showed significant enrichment of nucleic acid metabolism proteins, thus suggesting a plausible protective role of methionine in gene regulation and cell division. This has been added in the revised manuscript (Fig. 8e-8g) and supporting information (Supplementary Fig. 35).

In-gel fluorescence analysis:

Proteomics analysis:

Gene ontology analysis:

Biological processes analysis

REVIEWERS' COMMENTS

Reviewer #1 (Remarks to the Author):

The revisions have addressed most of the primary concerns raised upon initial review, and publication now seems warranted.

Reviewer #3 (Remarks to the Author):

In this revised manuscript, Sadu et al. addressed all of my issues with the proposed experiments.

As a minor comment, while the observed amino acid selectivity clearly favors labeling of methionine, I would not simply disregard the modification on other amino acids simply on the basis that this was not observed on peptide/purified proteins as depending on the protein microenvironment these may very well be real in a complex system like cellular lysate or live cells.

In conclusion, I now support the publication of this manuscript in Nat Commun.

TITLE: Copper(I)-Nitrene Platform for Chemoproteomic Profiling of Methionine

We want to take this time to sincerely thank the reviewers for their insight into bettering our research. We have addressed the reviewers' concerns in a revised version of our manuscript. Please see our responses to the reviewers below.

Reviewer #1 (Remarks to the Author):

Comment: The revisions have addressed most of the primary concerns raised upon initial review, and publication now seems warranted.

Justification: We thank the reviewer for accepting our revisions.

Reviewer #3 (Remarks to the Author):

In this revised manuscript, Sadu et al. addressed all of my issues with the proposed experiments.

Comment: As a minor comment, while the observed amino acid selectivity clearly favors labeling of methionine, I would not simply disregard the modification on other amino acids simply on the basis that this was not observed on peptide/purified proteins as depending on the protein microenvironment these may very well be real in a complex system like cellular lysate or live cells.

In conclusion, I now support the publication of this manuscript in Nat Commun.

Justification: We appreciate your insightful comment and agree with your perspective. We acknowledge that the modification of other amino acids, although not predominant in our peptide/purified protein experiments, could indeed occur under the complex conditions present in cellular lysates or live cells. This possibility will be highlighted as a consideration for future studies and in the interpretation of our results in more heterogeneous environments. Thank you for bringing this important point to our attention.